# Nuclear export of the pre-60S ribosomal subunit through single nuclear pores observed in real time

Jan Andreas Ruland [1], Annika Marie Krüger[1], Kerstin Dörner[2], Rohan Bhatia[1], Sabine Wirths[1], Daniel Poetes[1], Ulrike Kutay [2], Jan Peter Siebrasse[1] & Ulrich Kubitscheck [1✉]

Ribosomal biogenesis has been studied by biochemical, genetic and electron microscopic approaches, but live cell data on the in vivo kinetics are still missing. Here we analyse the export kinetics of the large ribosomal subunit (pre-60S particle) through single NPCs in human cells. We established a stable cell line co-expressing Halo-tagged eIF6 and GFP-fused NTF2 to simultaneously label pre-60S particles and NPCs, respectively. By combining single molecule tracking and super resolution confocal microscopy we visualize the dynamics of single pre-60S particles during export through single NPCs. For export events, maximum particle accumulation is found in the centre of the pore, while unsuccessful export terminates within the nuclear basket. The export has a single rate limiting step and a duration of ~24 milliseconds. Only about 1/3 of attempted export events are successful. Our results show that the mass flux through a single NPC can reach up to ~125 MDa·s$^{-1}$ in vivo.

[1] Institute of Physical and Theoretical Chemistry, University of Bonn, Wegelerstr. 12, 53115 Bonn, Germany. [2] Institute of Biochemistry, ETH Zurich, Otto-Stern-Weg 3, 8093 Zurich, Switzerland. ✉email: u.kubitscheck@uni-bonn.de

Ribosome biosynthesis is a cellular mammoth task requiring the orchestrated interplay of >200 proteins over three different compartments in the eukaryotic cell. Mature ribosomes contain four ribosomal RNAs (rRNA) and ~80 proteins and are built from the small 40 S and the large 60 S subunit. The biogenesis of the subunits is a complex process and begins with the synthesis of the rRNAs, which provide the backbone for both subunits (for review, see refs. [1,2]). In eukaryotic cells, three of the four rRNAs are transcribed by RNA polymerase I (Pol I) as one single precursor in the nucleolus, while the 5 S rRNA is transcribed by Pol III. Pre-rRNA synthesis, cleavage and maturation go along with the integration of ribosomal proteins and of multiple accessory factors and finally results in the small pre-40S and a large pre-60S subunit. The eukaryotic pre-60S particle (MW 2.1–3.1 MDa)[3,4] is ~25 nm in diameter while the pre-40S subunit has an average diameter of 16 nm. The subunits are among the bulkiest RNA transport substrates that have to be exported out of the nucleus.

The export pathways of the different RNA classes can be distinguished by the transport receptors used and how they are loaded to the RNA (see refs.,[5–7] and refs. therein). Small RNA molecules like transfer RNA, micro RNA or small nuclear RNAs are exported by 'importin β-like' receptors, e.g., Exportin-t or Exportin-5 (Exp5), which are bound directly in a RanGTP-dependent manner. In contrast, rRNA and messenger RNA (mRNA) are exported as intricate ribonucleoprotein (RNP) particles. The export of mRNA requires Mex67/Mtr2 in yeast and Tap/p15 (or NXF1/Nxt1) in humans and their mRNP incorporation is mediated by the adaptor proteins. The directionality of the mRNA export does not rely on the RanGTP/GDP gradient. Rather, an ATP-dependent RNA helicase, DDX19 in mammals (Dpb5 in yeast), together with additional factors remodels and releases the mRNPs after export into the cytoplasm[8,9].

In yeast, Mex67/Mtr2 is also involved in the export of the ribosomal subunits[10] but acts in doing so together with additional export factors, i.e., Rrp12p[11], Arx1[12,13], Ecm1[14], Bud20[15], Npl3[16] and Gle2[17] for the pre-60S particle (for review, see ref. [7]). The Mex67/Mtr2 heterodimer is bound to the late pre-60S subunit by direct interaction of its loop insertions in the middle NTF2-like domains of both Mex67 and Mtr2 with the 5S-rRNA[10]. Tap/p15 does not have these insertions and thus does not play a role in ribosomal export. Instead, in human cells, Exp5 is also needed for pre-60S export[18].

Although ribosomes are evolutionary highly conserved the only transport receptor unambiguously identified in the nuclear export of both the yeast and the human pre-60S particle is Xpo1/Crm1, which recognises the nuclear export signal (NES) of ribosome-bound NMD3 in a RanGTP-dependent manner[19–21]. After export, the Crm1 and Exp5 are removed by RanGAP induced GTP hydrolysis on Ran (reviewed in ref. [22]).

The most-detailed view of the actual pore transit of pre-60S particles derives from a recent electron tomographic study[23]. These researchers analysed NPC passage of the pre-60S particle in snap-frozen yeast cells. They did not label the subunits but identified them in transit through NPCs by their characteristic size and morphology. It was shown that the pre-60S particles move through the central channel of the NPC and that ~4–5% of all yeast NPCs contain a subunit in transit at any given time point. Of course, electron microscopy studies can only provide a static view of a dynamic process but using a skilful probabilistic approach Delavoie et al. estimated a translocation time of ~90 ms.

So far, live-cell data on the ribosomal export are missing. However, pre-40S and pre-60S particles represent large RNP particles that can be labelled by a comparable approach as we have used previously for transport experiments on mRNA particles[24]. Only recently, we succeeded in labelling the pre-40S particles using DIM2 (Pno1) coupled C-terminally to a SnapTag[25]. Our approach enabled us to observe pre-40S particles inside cell nuclei at the single-particle level. Here, we used a similar approach to visualise pre-60S particles by employing eIF6, which is loaded onto the pre-60S ribosomal subunit in the nucleolus. Already some years ago it was demonstrated that eIF6-HaloTag is functionally incorporated into pre-60S particles and represents a feasible approach to fluorescently label pre-60S particles in vivo[26]. Here, we established a HeLa cell line stably expressing eGFP-tagged nuclear transport factor 2 (NTF2)[27], a transport receptor highly enriched within NPCs in order to precisely localise single nuclear pore complexes. This cell line was re-transduced to additionally express eIF6-HaloTag, which was subsequently labelled with the membrane-permeable dye JF549[28]. By adding JF549 at sub-nanomolar concentrations, we succeeded to optically singularise pre-60S particles, whose trajectories could well be traced by single-particle tracking. The quasi-simultaneous observation of single NPCs by super-resolution confocal laser scanning microscopy enabled the real-time observation of nuclear pre-60S particle export. From the single-particle data, we could deduce the export dwell time, locate the route of successful and unsuccessful export events and identify a rate-limiting step.

## Results

**eIF6-Halo-tagged pre-60S particles are functional.** In order to label the pre-60S particles, we targeted eIF6, which is stably incorporated into the particle[29] and prevents its premature association with the pre-40S subunit. We created a HeLa cell line stably expressing both eGFP-NTF2 and eIF6-HaloTag. The HaloTag allows for labelling with photostable dyes such as JF549 and to visualise single pre-60S particles by adjusting the dye concentration. Staining eIF6-HaloTag by JF549 at a concentration of 0.1 μM resulted in the typical nucleolar accumulation of ribosomal subunits (Fig. 1a and Supplementary Fig. 1) and suggested that the Halo-tagged eIF6 exists preferentially bound to pre-60S particles as was previously concluded by Gallo et al.[26].

To further test whether the tagged eIF6 acts like the wild-type protein, we performed pulldown experiments using the HaloTag. Immunoblotting revealed that the purified complex contained Rpl10 and confirmed that the tagged eIF6 is successfully assembled into the subunit (Fig. 1d).

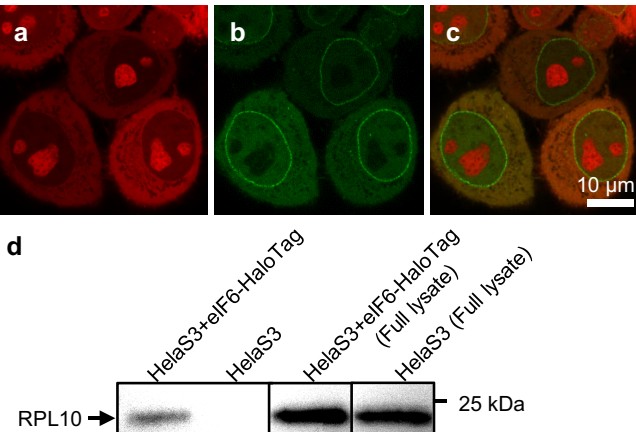

**Fig. 1 HeLa S3 cells expressing eGFP-NTF2 (green) and eIF6-HaloTag (red). a** eIF6-HaloTag-JF549 staining. **b** eGFP-NTF2. **c** Overlay of both channels. **d** Immunoblotting against Rpl10 after pulldown of eIF6-HaloTag in HeLa S3 cells stably expressing eIF6-HaloTag and untreated HeLa S3 cells. Loading controls are shown on the right-hand side.

The biochemical and microscopic data were also consistent with the result of incubation of the eIF6-JF549 cells with the transcriptional inhibitor Actinomycin D. RNA polymerase I catalysing ribosomal RNA transcription is most sensitive to the drug[30], which abolishes ribosome biogenesis and causes a redistribution of nucleolar components including nascent pre-ribosomes[31]. We observed a complete loss of nucleolar labelling, suggesting incorporation of eIF6-HaloTag into pre-60S particles (Supplementary Fig. 2).

To test if labelled pre-60S particles can be exported successfully, we treated the cells with Leptomycin B (LMB). LMB prevents binding of the export receptor Crm1 to NMD3 and thus blocks the nuclear export of pre-60S particles[20,32]. Correspondingly, we observed an increase in intranuclear fluorescence when we applied LMB to HeLa cells expressing eIF6-HaloTag molecules, which were marked by JF549 (Supplementary Figs. 3a and 3b). The fluorescence increase was clearly detectable, but not exhaustive (Supplementary Fig. 3c). We suspect this was due to a fraction of unbound eIF6-HaloTag-JF549 molecules. This view was also supported by a determination of the fraction of eIF6-HaloTag that was pre-60S particle-associated by western blotting for eIF6 across a sucrose gradient (Supplementary Fig. 3d).

**Tracking of single pre-60S particles at single NPCs**. By adjusting the JF549 concentration between 0.3 and 1 nM we could visualise single pre-60S particles in live cell measurements. In order to directly observe the intracellular transport of single pre-60S particles, we combined a narrow-field HILO illumination[33] and EMCCD detection with an LSM880 Airyscan in a customised microscopic set up. This enabled us to acquire movies of single pre-60S particles using the EMCCD detection path and parallel detection of GFP-tagged NPCs revealing the position of the nuclear envelope (NE) in the very same cell using confocal Airyscan imaging (see Fig. 2 and Movie S1). The particles were mostly immobile inside nucleoli, but in the nucleoplasm-diffusive motion was prevailing. Tracking of the single particles revealed a maximal nuclear diffusion coefficient of $1.7 \pm 0.1$ µm²/s (see Methods). The corresponding jump distance distribution indicated also a small fraction of significantly faster diffusing particles. Presumably, these jumps were due to free eIF6-HaloTag-JF549 molecules. The value of $1.7 \pm 0.1$ µm²/s was smaller than the maximal diffusion coefficient that we recently determined for single pre-40S particles, $2.3 \pm 0.3$ µm²/s[25], and significantly smaller than the maximal diffusion coefficient of free, unbound proteins in cell nuclei, >10 µm²/s[34]. Together with the results of the bulk experiments reported above, this showed that the respective particles corresponded to single pre-60S particles.

HeLa cells comprise ≥3000 NPCs in their NE[35–37], and display nearest neighbour distances of ~130 nm[38]. Our LSM880 Airyscan achieved optical super-resolution (~150 nm laterally) in the green fluorescence channel using an ×63 NA 1.46 objective lens. Thus, with this instrument, we were able in many cases to identify single NPCs (Figs. 2 and 3). Control measurements revealed that we could determine the position of single NPCs with a precision of <1 nm over a time period of 20 s (Supplementary Fig. 4).

In order to correlate the observed pre-60S particle positions to locations within the cell nuclei and especially the NE, a precise alignment of the images acquired with the two different detectors was compulsory. To this end, we used UV fluorescent microbeads, which were immobilised on the cell surface of the HeLa cells (see Methods). They served as reference markers that could be observed by both the EMCCD and the LSM880. The corresponding point signals were used to register the two fluorescence channels to each other (Fig. 2b and Supplementary Fig. 5).

After the image registration process, we could observe single pre-60S particles diffusing within cell nuclei, the NE of which was specified by a line of eGFP-NTF2 loaded NPCs (Fig. 2c). Single subunits could be seen approaching and touching single NPCs. Often, we observed subunits probing one or several NPCs – until finally they were exported into the cytoplasm (Fig. 3a–f, Movie S2). In many cases, however, the contact with an NPC did not result in an export, but particles rather returned into the nucleoplasm.

We defined a subunit as interacting with an NPC, when the distance between the subunit location and the NPC position was smaller or equal than a certain threshold, $d_{int}$. The value for $d_{int}$ was deduced from the geometrical extension of the NPC and the localisation precision of the respective detection paths, and their combined co-localisation precision. This resulted in $d_{int} = 129$ nm (for details, see Methods).

In order to achieve a quantitative evaluation of the pre-60S particle–NE interactions from the tracking data and the transformed membrane image of each cell, we developed a fully automated analysis pipeline comprising three major sections. First, the position of single NPCs in the green channel was automatically determined (see Supplementary Fig. 6) returning a list with coordinates of single NPCs. Next, NPC coordinates were compared with the single-particle tracking results. Thus, we identified for every individual NPC all tracks of interacting pre-60S particles (see Supplementary Fig. 7). We would like to stress that the trajectories identified in this manner did not correspond to all particles, which interacted with the NE, because the NPC list contained only those NPCs, which were clearly singularised and thus identifiable, i.e., was located sufficiently far away from other NPC signals.

Next, the list containing the NPC coordinates was used to create a polygon, which roughly described the shape of the cell nucleus (see Supplementary Fig. 8). This polygon was used by a final algorithm that defined the exact type of interaction of the particle with the NPC, i.e., to decide about, e.g., rejected or successful transport events and the direction of transport (Fig. 3g, Supplementary Fig. 9). Often, not all steps of the interaction with the NPC could be recorded, e.g., because the observed particle came from or vanished into out-of-focus regions. All observations of interactions with NPCs were sorted into categories by discriminating among all possible interaction types for particles moving near the NE (Fig. 3g). In this manner, movies from 230 different cells with an overall run time of ~90 mins were evaluated.

**Nuclear export of pre-60S particles**. Altogether, we identified 78 complete export events, for which we could see the particle approaching from the nucleoplasm, interacting with an NPC, and finally dissociating into the cytoplasm. For each export trajectory, the interaction duration with the translocating NPC was calculated. A histogram of the translocation times is shown in Fig. 4a. From all export events, we determined a mean translocation time of $35 \pm 20$ ms. The export dwell time kinetics were obtained by fitting an exponential decay model to the probability distribution of the individual translocation times. To test if different populations of particles with different translocation times exist, the dwell time distributions were modelled by single, double or triple-modal functions (for details, see Methods). The Akaike Information Criterion[39,40] was used to evaluate the three models. This yielded unambiguously that only a single-particle species with a single translocation duration existed. The analysis indicated the existence of a single rate-limiting step. In this manner, we determined $\tau = 24 \pm 4$ ms as the expectation value for the export duration.

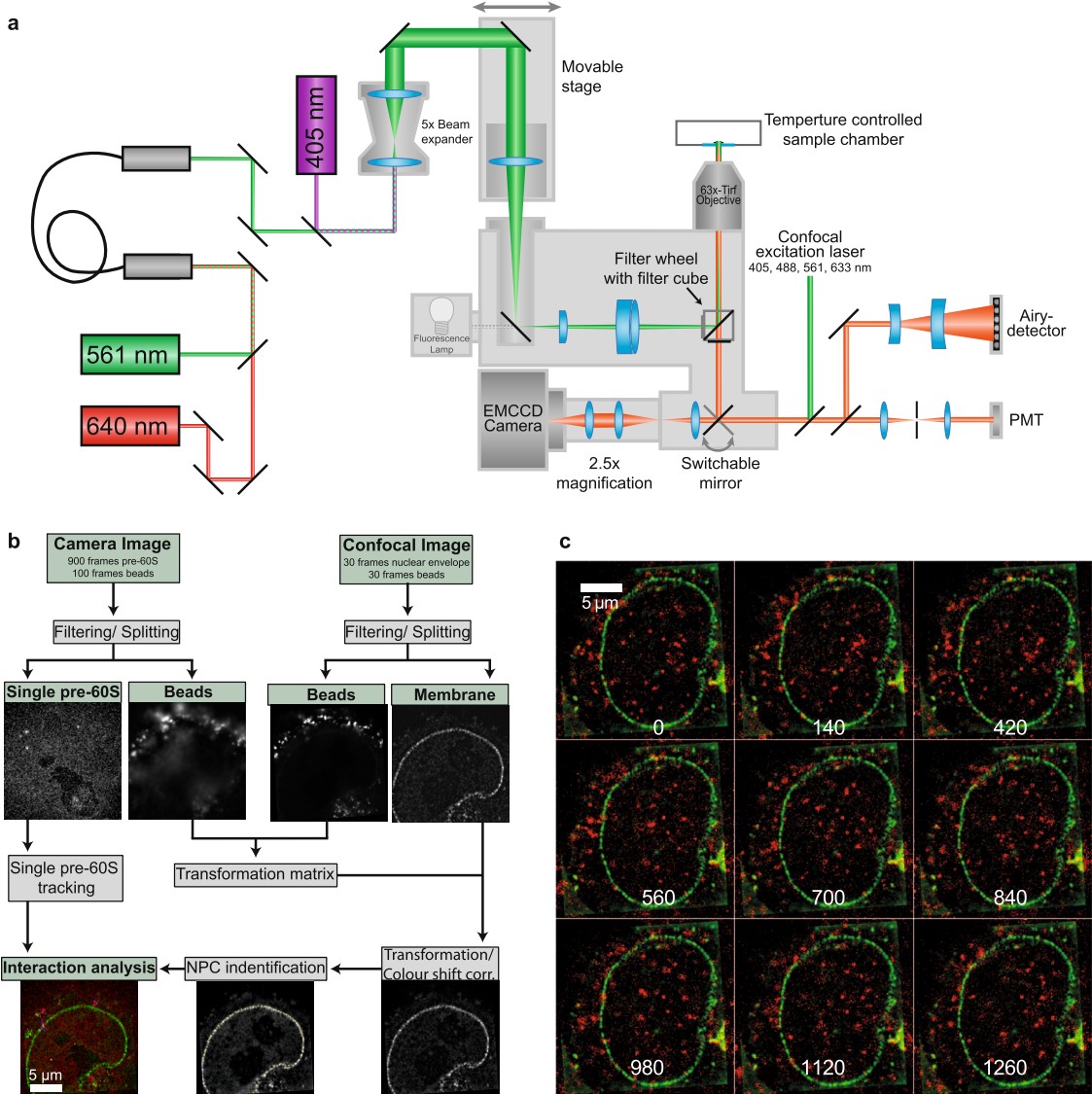

**Fig. 2 Single-molecule and super-resolution confocal microscopy of the same cell. a** A Zeiss LSM880 was extended by an additional beam path for narrow-field single-molecule excitation and high sensitivity detection. This beam path was used for imaging single pre-60S particles labelled by eIF6-HaloTag-JF549 (red). By switching the fully automated microscope to confocal super-resolution mode we were able to image the very same sample region by Airyscan microscopy, which revealed single NPCs lined up at the NE labelled by eGFP-NTF2 (green). For image registration, UV fluorescent microbeads coupled to the cell surface, which were seen in both modes, were used. **b** Sketch of the measurement principle. Using laser illumination with a wavelength of 561 nm a movie of single eIF6-HaloTag-JF549 (red) inside a cell nucleus was acquired at a high frame rate. After movie acquisition illumination switched from 561 to 405 nm to acquire images of the reference beads. Subsequently, super-resolution imaging of the NPCs labelled by eGFP-NTF2 (green) and of the reference beads was achieved by Airyscan microscopy in the green and UV fluorescence channels, respectively. The EMCCD movie was used to determine pre-60S particle tracks in the nucleus and across the NE. The super-resolution images of the NTF2-labelled NPCs were employed to identify single NPCs and their positions along the NE. The two sets of the reference bead images were used to calculate the transformation matrix that was needed to map the pre-60S trajectories onto the NPC positions. **c** Hela cells stably expressing NTF2-eGFP (green) and eIF6-HaloTag labelled by JF549 (red) after image registration. Single pre-60S particles could be discerned. Numbers represent time in ms.

In order to extract information on the spatial position of the rate-limiting step of the transport process, we aligned all complete export trajectories such that the position of the transporting NPC was defined as the origin of a coordinate system and the transport direction from nucleoplasm to cytoplasm coincided with the positive $x$ axis (Supplementary Figs. 10 and 11). There are various options to accomplish this, we decided to use the two nearest neighbour NPCs as defining the direction of export (for details, see Methods and Supplementary Fig. 11). Then, we plotted all export trajectories into this new coordinate system (Fig. 4b, top).

In order to determine the spatial position of the rate-limiting step of the transport process, we divided the transport axis into bins of 50 nm and counted the trajectory positions in the respective bins considering all particles that were observed within ±100 nm of the pore axis in $y$ direction. For this analysis, we used the point of maximum NTF2 fluorescence intensity as marker for the NPC location. The location of the maximum NTF2 fluorescence was related to the NPC topology using mAb414 labelling Nup 62 and various other nucleoporins[41] as reference (Supplementary Fig. 10 and Methods). Thereby, we found that NTF2 binds preferentially between 0 and 70 nm off the central

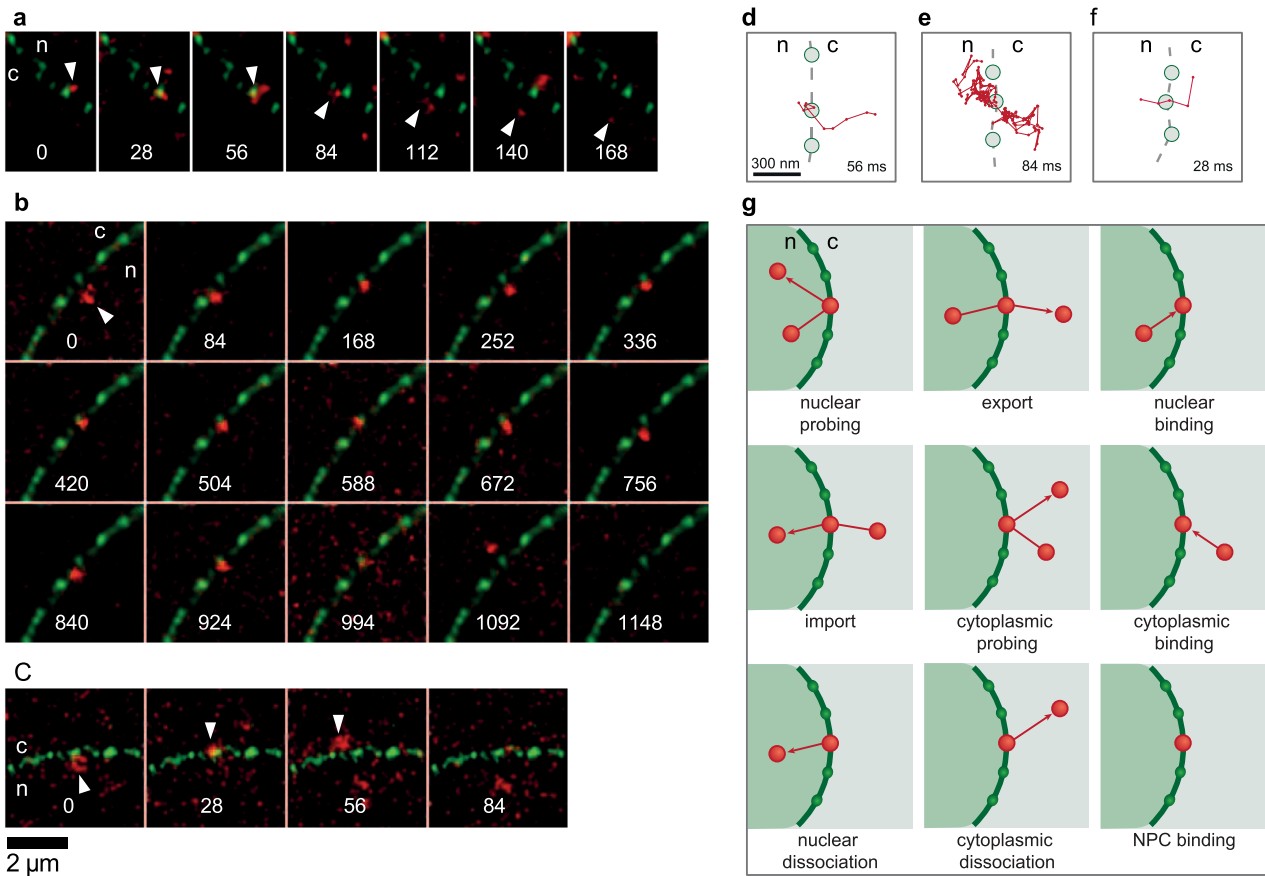

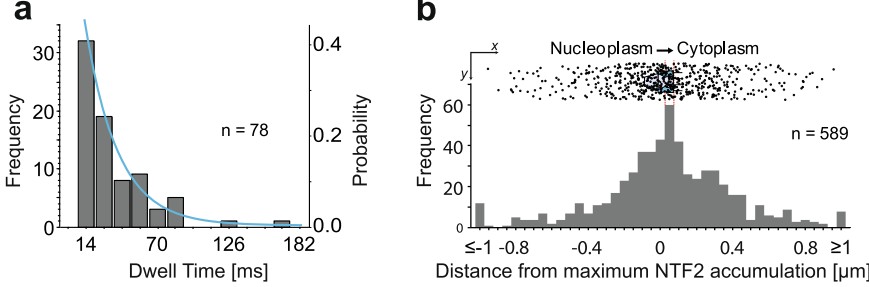

**Fig. 3 Single pre-60S particles (red) passing a single NPC (green). a–c** Image sequences showing export events. The numbers represent time in ms. **d–f** Trajectories (red) of the particles shown in **a–c** rotated such that the transport direction is parallel to the x axis, n nucleoplasm, c cytoplasm. Also, the position of the central and two neighbouring NPCs were shown. The given times correspond to the respective export dwell times. **g** All possible ways, in which the pre-60S particles could interact with the NPCs.

**Fig. 4 Analysis of pre-60S export events. a** Distribution of dwell times at the pore. The bar height indicates the frequency of the observation of the respective dwell times. The blue line indicates the best fit of the probability distribution using an exponential decay model with one translocating species yielding an export duration of $\tau = 24 \pm 4$ ms. The total number of evaluated trajectories was $n = 78$. **b** Superposition of all export trajectories aligned to the transporting NPC. (Top) The trajectories were aligned to the transporting NPC (as indicated by the maximum of the eGFP-NTF2 signal) as the origin and then rotated such that the export direction coincided with the positive x axis. (Bottom) The number of pre-60S particle positions in a y axis region of ±100 nm was plotted in dependence on the location with regard to the average NTF2 accumulation. The maximum occurred at the position of the cytoplasmic ring of the NPC (red lines). The total number of single positions shown is $n = 589$.

NPC plane with its maximum at 29 nm towards the nuclear basket. This binding site distribution was in accordance with previous data[42–45].

The resulting binding site distribution of the pre-60S particles and a corresponding sketch of the NPC structure indicate that the translocating particles were most often observed in the region of the cytoplasmic ring of the NPC (Fig. 4b). We assume that the passage of this position corresponded to the rate-limiting step of the transport process.

As mentioned, the analysis pipeline detected $n_{exp} = 78$ reliable export events. This number might seem to be small. However, it was expected that $n_{exp}$ would be relatively low, because a number of strict preconditions had to be fulfilled for an event to be counted. The identification of an export event required that the translocating particle remained in focus during the NE approach, the NPC translocation and dissociation into the cytoplasm. In the 3D space of the cell, diffusive motion away from the focal plane is simply much more probable[24]. Furthermore, our algorithms

| Table 1 Number of detected interaction events with single NPCs sorted according to the type of interaction. | | | | | |
|---|---|---|---|---|---|
| Category | No. of events | Category | No. of events | Category | No. of events |
| Nuclear probing | 145 | Export | 78 | Nuclear binding | 552 |
| Import | 2 | Cytoplasmic probing | 761 | Cytoplasmic binding | 568 |
| Nuclear dissociation | 229 | Cytoplasmic dissociation | 532 | NPC binding | 799 |
| Undefined | 683 | | | Total | 4349 |

required a clear and distinct identification of the transporting NPC. In certain cases, we could observe a translocation process, but the corresponding NPC could not unambiguously be identified, e.g., owing to a high surface density of NPCs in the respective region of the NE. In addition, the category 'cytoplasmic dissociation' (Fig. 3g and Table 1) contained events that presumably corresponded to export events, but their first association step with the NPC could not be detected. Similarly, there were numerous approaches to NPCs from the nucleoplasmic side ('nuclear binding', $n = 552$), for which we did not observe the release from the pore. Presumably, quite a fraction of these resulted in export, whose final step we missed. Also, there were ~800 cases, for which we observed the pre-60S particles only at NPCs and did neither observe the approach nor a release ('NPC binding'). Finally, a valid observation not only required a clearly recognisable line of NPCs in focus and a certain density of particles in the observation field, but also the presence of a sufficient number of UV reference beads in the respective focal observation plane. Fulfillment of all these prerequisites was needed to obtain useful transport data.

**Abortive export events**. In addition to the complete observed export events, our analysis also returned attempted or abortive export events in the category 'nuclear probing' (Fig. 3g). These were defined by a particle approach to the NE from the nucleoplasm, an interaction with a specific NPC, and then return back into the nucleoplasm. Interestingly, in this category, we observed more events, namely 145 (Table 1). Obviously, also the category 'nucleoplasmic dissociation' comprising 229 events that described failed or interrupted export events.

Again, we determined the interaction duration of the failed export events with the NPCs and the distribution of binding sites along the NPC symmetry axis (Fig. 5). Not surprisingly, the interaction duration of the subunits, whose export failed, was shorter than that of completed exports, namely only $\tau = 13 \pm 3$ ms.

**Interaction of single pre-60S particles with NPCs in the presence of LMB**. Finally, we examined the nuclear export of pre-60S particles upon addition of LMB, which inhibits export as discussed above. As expected, the observation of export events was reduced strongly from 0.91 events/min to 0.13 events/min 120 min after the addition of LMB and thus became very rare (Supplementary Fig. 12a). This inhibition confirmed that the observed nuclear export events (Figs. 3 and 4) were due to eIF6-HaloTag bound to pre-60S particles. Interestingly, the distribution of binding sites for the failed export events did not significantly differ for trajectories observed in the presence or absence of LMB (Supplementary Figs. 12b and 12c).

**Discussion**
Here, we report on the nuclear export of single pre-ribosomal particles through NPCs by live microscopy. The analysis of the transport process kinetics and of the binding site distribution of the pre-60S particles along the transport axis of NPCs allowed us

to define key steps of the export process and the position of the rate-limiting step of the translocation through the central NPC channel.

In order to visualise single pre-60S particles, we used an indirect fluorescence-labelling approach. We stably expressed eIF6-HaloTag in HeLa cells and labelled it in vivo by the fluorescence dye JF549. Using extremely low fluorophore concentrations it was possible to observe single, mostly mobile, diffraction-limited spots in live cells by sensitive fluorescence microscopy. We provided fivefold evidence that these spots represented pre-60S particles. In bulk experiments, the fluorescent particles showed a nucleolar accumulation mirroring the localisation of tagged ribosomal proteins, such as RPL29, in pre-60S reporter cell lines[18]. The addition of the transcriptional inhibitor actinomycin D resulted in a loss of the typical nucleolar fluorescence, presumably caused by a stop of the production of pre-ribosomal particles. A pulldown experiment demonstrated that the labelled eIF6 was contained in molecular aggregates that comprised also the ribosomal protein Rpl10, an established component of pre-60S particles(reviewed by[46]). Incubation with LMB, an inhibitor of pre-60S nuclear export by blocking Crm1, resulted accordingly in an increase of nuclear fluorescence. Finally, the observed single particles diffused more slowly than the smaller pre-40S particles and drastically more slowly than unbound proteins. Altogether this suggested that eIF6-HaloTag-JF549 was contained in pre-60S particles and allowed us to visualise pre-60S particle dynamics in living cells.

We used super-resolution confocal laser scanning microscopy of eGFP-NTF2-expressing HeLa cells to reveal the location of the NE. Distinct and characteristic fluorescence maxima along the NE indicated the positions of single NPCs. Quasi-simultaneously we imaged the nuclear transport of the pre-60S subunits in the same cellular region. In this way, we were able to observe pre-60S particles interacting with single NPCs. These events were analysed by an elaborate automatic data processing pipeline.

Altogether, we identified 78 complete export events for which we monitored an approach from the nuclear side to the NPC, an encounter with the NPC and a release into the cytoplasm. The data analysis extracted the transport trajectories during this process revealing interaction durations (dwell times) with the NPCs and the respective binding positions. The distribution of the dwell times at the NPCs could best be described by a mono-exponential decay function as was shown by the analysis of the distribution using the Akaike information criterion yielding an expectation value for the translocation duration of $24 \pm 4$ ms.

This value is in very good agreement with estimations for the maximum transport rate of cargo molecules through the human NPC formerly estimated by Ribbeck and Görlich[35,47]. Based on their bulk measurements they assumed that ~1000 importin β per second can translocated through a single NPC, which equates to a mass flux of ~100 MDa s$^{-1}$. From the electron tomography study of Delavoie et al., it is evident that ribosomal subunits are exported one at a time and not in parallel. The translocation duration of $24 \pm 4$ ms in principle would allow the transport of ~35–50 large subunits per second through a single pore. Assuming an average molecular mass of ~2.5 MDa for the human pre-60S

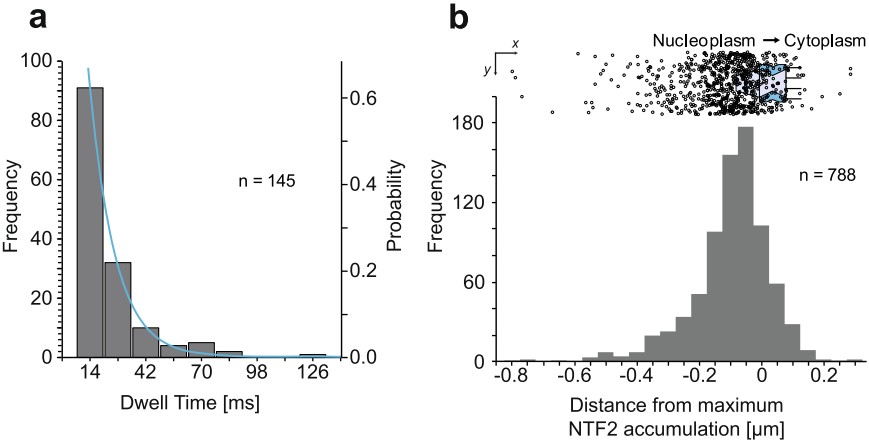

**Fig. 5 Failed nuclear export of pre-60S particles. a** The bar height indicates the frequency of the observation of the respective dwell times. The total number of evaluated trajectories was $n = 145$. The blue line indicates the best fit using a single exponential decay function and yielded $\tau = 13 \pm 3$ ms. **b** (Top) The trajectories were aligned to the transporting NPC (as indicated by the maximum of the eGFP-NTF2 signal) as the origin and then rotated such that the export direction coincided with the positive $x$ axis. (Bottom) The number of pre-60S particle positions in a $y$ axis region ±100 nm off the central NPC was plotted as a function of the location with regard to the average NTF2 accumulation. The maximum clearly coincides with the nuclear basket. The total number of single positions shown is $n = 788$.

particle this accounts for a mass flux of 87.5–125 MDa s$^{-1}$ per NPC or ~875–1250 importin β s$^{-1}$ per NPC. Our live-cell data show that the maximum transport rate is almost reached in vivo and is higher than the suggested in vivo minimum rate of 10–40 MDa s$^{-1}$ for a growing cell[22,47].

Notably, the measured translocation duration of $24 \pm 4$ ms is in the same order of magnitude as the ~$90 \pm 50$ ms for the dwell time of pre-60S particles in the yeast NPC, which was estimated by Delavoie et al.[23]. These authors observed pre-60S particles within NPCs in EM tomograms and used a probabilistic queueing model to extract a transport dwell time from the NPC occupancy. Among other parameters their estimate was based on the number of NPCs available for transport. Smaller NPC numbers would lead to shorter translocation times. In our experiments, we observed—apart from complete export events—also a large fraction of attempted or failed export events. These unsuccessful events might indicate that numerous NPCs are temporarily 'occupied', e.g., with other bulky substrates and indeed not accessible for ribosomal export. This would effectively reduce the number of NPCs and lead to a lower translocation time estimate in the queueing model. Therefore, we consider our result as a support of their model.

Ribbeck and Görlich[47] found that the transport rates of bigger transport cargoes scale with the number of loaded transport receptors. This was later confirmed by studies of nuclear import of exceptionally big artificial transport substrates in permeabilized mammalian cells[48–50]. Lowe et al.[48] used quantum dots (QD) functionalized with importin β-binding domains featuring a hydrodynamic radius of ~18 nm. Reducing the number of bound importin β molecules dramatically increased the QD dwell time in the central NPC channel[48]. Tu et al.[49] engineered β-galactosidase (β-gal) to carry four M9-signals resulting in an import cargo of comparable size ($18 \times 15 \times 9$ nm³). Their single-molecule analysis revealed that loading the β-gal with one receptor allowed NPC binding but no translocation. For maximum import efficiency loading of four transportins per β-gal was necessary. In a more recent study[50], the import of artificial virus capsids with increasing diameters (17–36 nm) and varying numbers of NLS (0–240) was quantified in ensemble measurements. Again, the initial transport rate and efficiency scaled with the number of NLS/transport receptors attached to the capsids.

Together, these studies illustrate that the enthalpic and entropic costs of solving a large cargo in the dense FG network can be compensated by multiple NTR-FG interactions.

In yeast, at least eight different transport receptors are loaded on a single pre-60S particle, i.e., Mex67/Mtr2, Rrp12p, Arx1, Ecm1, Bud20, Npl3, Gle2 and Xpo1, but for the human pre-60S particle so far only two main export receptors are known: Crm1 and Exp5. Only one Crm1 can be loaded to the NES of NMD3. Exp5 can recognise structurally diverse RNAs[22] and recognises its rRNA cargo in a sequence-independent manner via double-stranded stem-loops[18]. It is tempting to speculate that in this way eventually more than one Exp5 is loaded on a single human pre-60S particle to ensure the necessary 'receptor density' or avidity[49] for such a bulky cargo.

After the translocation, the cargo/receptor complex is dissociated on the cytoplasmic side of the NPC. Mex67/Mtr2 is removed by the ATP-dependent helicase activity of Dbp5 from mRNA, but surprisingly this does not occur for pre-ribosomal export[51]. For the RanGTPase-dependent receptors Exp5 and Crm1 engaged in human pre-60S particle export it is known that they are released after RanBP1/RanGAP1-triggered GTP hydrolysis. For Crm1-mediated export, nucleoporin Nup214 has been shown to play a critical role in this final step[52,53]. Moreover, Nup214 (Nup159 in yeast) is also critical for the pre-60S particle export in mammalian[18] and yeast cells[54]. RanGAP cannot directly act on Ran in the export complex, but needs RanBP1 or the Ran-binding domains of RanBP2/Nup358[22]. Nup214, Nup88 and Nup358 are the only nucleoporins located asymmetrically on the mammalian NPC and in human cells the Nup214/Nup88 sub-complex mediates the attachment of Nup358 to the NPC[55]. Our finding of a single rate-limiting step and single maximum of binding sides in the central NPC confirms a model, where both Exp5 and Crm1 are needed for NPC entry and passage of the pre-60S particle, but only the Nup214/Crm1 interaction is needed for release from the NPC[22,23,56,57]. Here, the strong and direct interaction between Nup214/Crm1 would 'fish-out' the pre-60S export complex from the central NPC channel via FG-repeats and hand it over to NUP358, which results in GTP hydrolysis and subsequent cytosolic pre-60S release (Supplementary Fig. 13).

Our analysis of the binding site distribution of subunits along the translocation axis and the topology of NPCs (Fig. 4b)

confirmed the results of Delavoie et al. (2019) and allowed a comparison with mRNP export. Our measured distribution of trajectory positions, although less well resolved than the EM data, showed a clear, single maximum in the central region of the NPC shifted towards the cytoplasm. Delavoie et al. (2019) observed a relatively low fraction of pre-ribosomes in the region of the nuclear ring, a large fraction at the inner ring of the NPC, and a medium-scale fraction at the cytoplasmic ring of the NPC, i.e., their binding site distribution showed a single maximum and a shoulder near the cytoplasmic filaments[23]. In agreement with their data, we also did not see an accumulation of binding sites in the region of the nuclear basket. Delavoie et al. (2019) suspected therefore that the pre-60S particle-nuclear basket interactions are of very transient nature[23]. This was supported by our observation that the dwell time of 'nuclear probing' (13 ± 3 ms) was short compared to the actual translocation step (24 ± 4 ms). The reason might be the missing quality control step for pre-ribosomes at the nuclear basket[23,58,59], like it exists for mRNPs[60,61].

Interestingly, for mRNP export trajectories a complementary distribution with two maxima on the nucleoplasmic and cytoplasmic faces of the NPC were observed[59,62]. It was suspected that the maximum at the nuclear face is due to quality control processes and/or RNP reorganisation for the translocation, and the maximum at the cytoplasmic face due to the removal of the mRNP export factors by the RNA helicase Dbp5, which infers the directionality of mRNP export (reviewed by[63]). As already stated, such complex enzymatic processes do not occur for pre-ribosomes.

Our data analysis allowed us to discriminate between successful and failed transport events. In the latter case, the respective pre-60S particles also interacted with NPCs according to our definition. Interestingly, we observed this significantly more often ($n = 145$) than successful export events ($n = 78$). Thus, only ~35% of attempted export events were successful. Similar fractions have been reported for mRNPs[24,62], so the value of ~1/3 of successful export events may present a general rule for large substrates. The binding site distribution of the failed exports showed a distinct maximum in the region of the nuclear basket, just outside the nuclear ring of the NPC. This supported the view that the respective pre-60S particles did not enter the hydrophobic NPC interior, but were blocked from it. We speculate that this was either due to occupancy of the NPC by another large transport cargo or indicated that the particles still missed an export factor, possibly Crm1.

In conclusion, we showed that nuclear export of pre-60S particles takes 20–30 ms and that ~1/3 of attempted export trials are successful. The multi-step light microscopic approach introduced here is laborious, but allowed for the first time the observation of single pre-60S particles during their interaction with NPCs in vivo. For the future, it is important to increase the experimental throughput. In that case, it could be combined with the mutational or RNA interference analysis of the nuclear transport machinery to refine our understanding of the transport of large RNA cargoes such as pre-ribosomes through the NPC. A comparable approach to study the nuclear export of pre-40S particles appears straightforward and would extend the range of questions to be asked.

## Methods
### Cell lines and labelling
*Plasmid constructs.* The cDNA of eIF6 was amplified from the Vector EX-I0088-M50 (Genecopeia, Rockville, MD, USA). The eIF6-cDNA was then cloned into the Gateway-enabled Vector pQCXIH-Halo-GW the Gateway system (Thermo Fisher, Waltham, MA, USA). The NTF2-cDNA (NM_005796.3) was cloned into the Gateway-enabled Vector pQCXIP-eGFP-GW (Clonetech, Mountain View, CA, USA) using the Gateway system.

*Cell line.* HeLa cell lines were generated using the Retro-X™ Tet-One™ Inducible Expression System (Clonetech, Mountain View, CA, USA) according to manufacturer's instructions. Transfections were performed using Xfect Transfection Reagent (Clonetech). First, a cell line with the eGFP-NTF2 construct was created and further modified with the eIF6-HaloTag construct. All cell lines were cultured in Dulbecco's Modified Eagle Medium (DMEM, Sigma-Aldrich, St. Louis, MO, USA) supplemented with 10% fetal bovine serum (Sigma-Aldrich), 2 mM glutamine (Sigma-Aldrich), 100 U/mL penicillin (Sigma-Aldrich) and 100 µg/mL streptomycin (Sigma-Aldrich).

*Transient expression.* For transient expression of proteins, 2 days before the experiment cells were seeded in a 35-mm-µ-Dish (Ibidi, Gräfelfing, Germany). To each dish 250 µL cell suspension and 750 µL DMEM were added. On the day before the experiment cells were transfected with 2 µg Plasmid using jetPRIME (Polyplus, Illkirch, France) according to the manufacturer's instructions.

*Fluorescence labelling.* After 24 h prior to the experiment, cells were split and seeded in a 35-mm-µ-Dish (Ibidi, Gräfelfing, Germany). To each dish 500 µL cell suspension and 500 µL DMEM were added. On the day of the experiment, Janelia-Flour-549-HaloTag-Ligand (JF549, Janelia, Ashburn, VA, USA)[28] was added (final concentration 0.1 µM for bulk staining and 0.3–1 nM single-molecule staining) and incubated for 30 min at 37 °C/5% CO₂. Cells were washed three times with 2 mL phosphate-buffered saline (PBS) buffer, and 1 mL of media was added.

### Tests of eIF6-HaloTag
*Isolation of proteins and immunoblotting.* Cells were grown to 90% confluence in two 75 cm² cell culture flasks, washed with 20 mL cold PBS and scraped off in cold PBS. The pulldown was performed using the HaloTag Mammalian Pull-Down System (Promega, Madison, WI, USA) using the manufacturer's instructions.

Proteins were separated by electrophoresis in a 12% Sodium dodecyl sulfate–polyacrylamide gel and then transferred to an Immune-Blot polyvinylidene fluoride Membrane (Bio-Rad, Hercules, CA, USA) at 25 V and 0.3 A for 35 mins. Membranes were blocked overnight at 4 °C with 3% milk powder in TBS-T (tris-buffered saline, 0.1% Tween20). Next, membranes were incubated first with anti-RPL10 antibody (1:500 in 3% milk powder in TBS-T, RPL10 Monoclonal Antibody (OTI6B11), #TA807662 (Lot: #VD2974632), Thermo Fisher, Waltham, MA, USA) for 2 h, washed 3× for 10 min with TBS-T and incubated with hrp antibody (1:1000 in 3% milk powder in TBS-T, anti-goat IgG-HRP, #sc-2304 (Lot: #C221), Santa Cruz Biotechnology) for 2 h at RT. The blot was developed using 1-Step Ultra TMB Blotting Solution (Thermo Scientific) using manufacturer's instructions.

*Blocking pre-60S particle export using Leptomycin B (LMB).* Cells were seeded and stained as described above and buffered with 100 mM 4-(2-hydroxyethyl)-1-piperazineethanesulfonic acid (HEPES). LMB solution (10 µM in Ethanol, Enzo Life Sciences, Lörrach, Germany) was added to a final concentration of 50 nM or 100 nM to the sample and the measurement was started directly. Cells were imaged for 3 h while being incubated at 37 °C. Every 10 min an image was taken using a Laser Scanning Microscope 880 (LSM880) and Zen Black (Zeiss, version 2.3 SP1)., whereas the focus was stabilised with the definite focus system of the microscope (Carl Zeiss Microscopy Deutschland GmbH, Oberkochen, Germany).

*Blocking RNA transcription using actinomycin D.* Cells were seeded and stained as described above and buffered with 100 mM HEPES. 10 µL of a 1 mg/mL actinomycin D solution (final concentration 10 µg/mL, Sigma-Aldrich) was added and the measurement was started directly. Cells were imaged for 24 h while being incubated at 37 °C. Every 10 min an image was taken using the LSM880 with focus stabilisation (Zeiss).

*Sucrose gradient centrifugation.* For sucrose gradient centrifugation, HeLa cells expressing EIF6-HaloTag were treated with 100 µg/mL cycloheximide for 3 min before harvest. Cells were lysed with a 27 G needle in 10 mM Tris pH 7.5, 100 mM KCl, 10 mM MgCl₂, 1% TritonX-100, 1 mM DTT, 100 µg/mL cycloheximide, and protease inhibitors. In all, 600 µg cleared lysate was layered on a 10–45% sucrose gradient. After centrifugation using a TLS55 rotor (Beckman Coulter) at 55,000 rpm for 85 min, gradient fractions were precipitated with TCA and analysed by immunoblotting.

### Light microscopy
*Modification of reference beads.* To 950 µL MES buffer (100 mM 2-(N-morpholino) ethanesulfonic acid (MES), 500 mM NaCl, pH 6) 50 µL Fluoresbrite Carboxy BB .10 Micron Microspheres (2.6% solid solution, Polysciences Inc., Warrington, PA, USA) were added, briefly vortexed and sonicated for 5 min. In all, 7 mg EDAC (1-Ethyl-3-(3-dimethylaminopropyl)carbodiimid) and 14 mg Sulfo-NHS (N-hydroxysulfosuccinimide) were added. The sample was shaken for 15 min at 1500 × g on a ThermoMixer C (Eppendorf, Hamburg, Germany). Then, the solution was transferred into an Amicon Ultra-4 Ultracel-100K Tube (Merck, Darmstadt, Germany). In all, 3 mL PBS were added and the sample was centrifuged for 5–7 min at 600 × g. The previous washing step was repeated. The final volume was

adjusted to 500 μL. Modified beads could be stored at 4 °C for 8 h and were prepared fresh for every day of measurements.

*Coupling of reference beads.* Directly prior to the measurement, the modified beads were sonicated for 3 min and cells were washed with 2 mL of PBS. Next, 950 μL PBS and 50 μL modified beads were added to the μ-dishes and incubated for 20 min at 70 rpm and 37 °C (Ecotron incubator, Infors HT, Bottmingen, Switzerland). Cells were washed three times with 2 mL PBS and 1 mL fresh media and HEPES (final concentration 100 mM) were added. Each sample was measured for maximally 1 h.

*Super-resolution confocal and single-molecule microscopy.* Super-resolution confocal microscopy to image single nuclear pores was performed using a Zeiss LSM880 equipped with an Airyscan detector (Zeiss) run under Zen Black (Zeiss, version 2.3 SP1). Images were acquired using the α-Plan-Apochromat ×63/1.46 Oil Corr M27 objective (Zeiss). For single-molecule microscopy, the aperture stop and the filter slider in front of the fluorescence lamp were replaced by a custom-made filter holder, which was equipped with a quad-line optical filter (F73-410, AHF analysentechnik AG, Tübingen, Germany). For excitation, lasers emitting at 561 (Obis LS, Coherent, Dieburg, Germany) and 405 nm (LDM-XT laser series, Lasos, Jena, Germany) were used. The beams passed a 5× beam expander (Thorlabs, Newton, NJ, USA) and were focused into the back focal plane of the objective by an auxiliary lens with a focal length of 200 mm. The focusing lens and a mirror were mounted on a mobile stage to achieve highly inclined and laminated optical (HILO) sheet illumination[33]. Fluorescence imaging was achieved using the same objective as for the confocal measurements with an additional 2.5× pre-magnification (Zeiss) in front of an EMCCD camera (iXon + 897D, Andor, Belfast, UK), which was placed at the camera port of the Axiovert200 microscope stage (Zeiss) and run using SOLIS (Andor, version X-4603). A filter cube comprising respective notch filters was used as an emission filter (F66-04TN, AHF). Lasers were controlled by custom-written software. The microscope was equipped with a heating chamber (Ibidi). All in vivo experiments were performed at 37 °C.

*Data acquisition.* Confocal scans were used to focus the equatorial region of a cell nucleus using eGFP-NTF2 fluorescence. Next, 1000 frames were acquired by the EMCCD camera. From these, 900 frames were obtained using 561 nm laser excitation and 100 frames using 405 nm. The central 200 × 200 pixels of the 512 × 512 camera chip were read out to optimise the imaging frame rate. Exposure Time was set to 13 ms and frame transfer was activated, which yielded a kinetic cycle time of 14 ms. EM gain was set to 300. Next, 20 confocal Airyscan images were acquired with the excitation alternating between 405 nm (1% laser power) and 488 nm (1.5% Laser power). The pixel size was set to 0.04 μm and the focus was stabilised during the measurements by the definite focus system (Zeiss).

### Image processing and analysis
*Image processing of the EMCCD camera images.* Image processing pipeline was programmed using Fiji[64] (version 1.52p) macros and plug-ins (programmed in Eclipse IDE, Eclipse Foundation, Inc, version Photon release 4.8.0). Images were converted from 'Sif' to '8-bit Tiff'. Image stacks were split into 100 frames of the reference beads and 900 frames of single-molecule images (Supplementary Fig. 5). For the reference stack, the background was subtracted with a rolling ball radius of 10, for the single-molecule data the radius was set to 50. The contrast of both stacks was enhanced (0.3% saturated pixel) and both were flipped vertically. The maximum intensity projection of the reference bead image stack was calculated. All images were scaled to the size of the confocal images.

*Image processing of the confocal images.* The Airyscan images were processed (strength of the filter was set to 'auto') and converted to 'Tif'. The two channels were separated, the background was subtracted (rolling ball radius of 10) and the contrast-enhanced (0.3% saturated pixel, see Supplementary Fig. 5). The first five optical slices of the bead reference images were averaged and the contrast was again enhanced (0.35% saturated pixel). The first five optical slices of the membrane image were summed. Both projections were converted to 8-bit.

*Determination of the transformation matrix.* To register the Airyscan and camera images, first, an affine pre-transformation was applied, then a transformation matrix was determined from the bead images and applied to the membrane image. Finally, the colour shift was corrected.

*Affine pre-transformation.* The affine pre-transformation matrix was applied using the custom-modified StackReg plug-in for Fiji (ImageJ)[64,65] to the confocal average projection of the beads to correct for rotation, shear and roughly correct translation (see Supplementary Fig. 5). To obtain the pre-transformation matrix Fluoresbrite Carboxy BB .10 Micron Microspheres (Polysciences Inc.) were immobilised in 2% agarose (1:100 v/v), imaged and processed like samples. The sum of the camera image and the average of the confocal image were added to one stack and aligned by an affine transformation using StackReg. The success of this transformation was visually controlled. About 20 of the resulting transformation matrices were averaged to obtain one average affine transformation matrix.

*Translation transformation.* Next, the remaining translation shift, caused by a shift of the microscope stage during measurement, needed to be corrected. The sum of the bead image of the camera and the pre-transformed confocal image was added to one stack and transformed by translation using StackReg. The resulting transformation matrix represents the specific transformation matrix for this pair of data. The success of this transformation was checked visually.

*Colour shift correction.* As the alignment of confocal and camera images was performed using beads with emission maximum at 407 nm a further colour shift correction was needed to align the membrane with the single-molecule channel. A mixture of Fluoresbrite Carboxy BB. 10 Micron Microspheres and 0.2 μm Tetraspeck (Life Sciences) was immobilised in 2% agarose (1:100 v/v) and imaged, processed and registered like the samples. The 900 frames, which were imaged at 561 nm with the EMCCD camera and the first five frames of the confocal channel at 488 nm were averaged. Both images were added to one stack and aligned by an affine transformation using the StackReg plug-in. The success of this transformation was controlled visually. About 20 of the resulting transformation matrices were averaged to obtain one colour shift transformation matrix.

First, the pre-transformation matrix, then the translation matrix and last the colour shift matrix were applied to the sum of the membrane image.

*Particle tracking.* Single-molecule tracking was performed using the Trackmate plug-in for ImageJ[66]. For spot detection, the LoG-based detector was chosen. The parameter 'estimated blob diameter' was set to 0.5 μm and 'sub-pixel localisation' was activated. An automatic threshold was used for 'initial spot filtering'. For tracking the 'Simple LAP Tracker' was chosen. Gap closing was allowed with a maximum closing distance of 1 μm and a maximum frame gap of the three frames. Maximum linking distance was set to 1 μm.

*Determination of localisation precision.* The single-particle localisation precision $\sigma$ was estimated according to Mortensen et al. (2010) as[67]

$$\sigma = \sqrt{F\left[\left(\frac{4}{3}\right)^2 \frac{s^2 + \frac{a^2}{12}}{N} + \frac{8\pi b^2\left(s^2 + \frac{a^2}{12}\right)}{a^2 N^2}\right]}$$

where $F = 2$ designates the noise excess factor for the EMCCD camera, $a$ is the pixel size (102 nm), $s$ is the width of half maximum (388 ± 15 nm) and was measured with 0.1 μm Tetraspeck beads immobilised in 2% agarose. $N$ is the number of photons per signal and was estimated to be 100–300 and $b$ is the background in photons per pixel, which was ~$N$/100. For 100 photons this resulted in a localisation precision of 37 ± 6 nm. For our data analysis, we used the upper limit—a localisation precision of 43 nm—for the single-molecule measurements.

### Determination of co-localisation precision. A mixture of Fluoresbrite Carboxy BB .10 Micron Microspheres and 0.2 μm Tetraspeck were immobilised in 2% agarose (1:100 v/v), imaged, processed, aligned and colour shift corrected like the samples. The 900 frames, which were imaged at 561 nm excitation with the EMCCD camera and the first five frames of the confocal channel excited at 488 nm were averaged and added to one stack. The remaining shift between the bead positions was measured using the TrackMate plugin for ImageJ with the same settings as for the particle tracking. This resulted in a co-localisation precision of 35 ± 2 nm. We used the upper limit of this value—37 nm—in the further analysis.

### Data analysis
*Determination of NPC positions and pre-60S subunit–NPC interactions.* To determine the positions of single NPCs (Supplementary Fig. 6), the contrast of the transformed membrane image was enhanced (0.35% saturated pixel). An intensity threshold (≥160) was set and the resulting mask five times dilated and five times eroded. Remaining artefacts were cleaned up manually. The mask and the enhanced membrane image were combined (Image calculator: 'AND') and local maxima were detected (ImageJ - 'Find maxima'). Through all maxima found in this manner four one-dimensional Gaussian functions were fitted in steps of 45° (see Supplementary Fig. 6) over a length of 11 pixels, respectively. In case the $R^2$ values of all single fits were ≥0.95 the maximum positions determined by these fits were averaged and taken as NPC location.

To define the outline of the NE, a polygon was created from all NPC positions (Supplementary Fig. 8). The centre of mass of all NPC positions of one cell nucleus was calculated and taken as a new origin. The NPC positions within this new coordinate system were transformed to polar coordinates. Then, the NPC positions were sorted by increasing polar angle $\phi$. In situations, where the different positions showed the same polar angle, they were sorted by smaller polar radius $r$.

To identify pre-60S particle tracks, which showed an interaction with single NPCs, we placed a virtual circle with a radius of 43 nm (the single-molecule localisation precision, see above) around every particle position of the tracks. Similarly, we placed virtual circles with a radius $r$ determined from the average geometrical pore radius ($r_p = 65$ nm), the single-molecule localisation precision ($\sigma_p = 43$ nm) and the co-localisation precision ($\sigma_c = 35$ nm) around all NPC

positions:

$$r \geq \sqrt{r_p^2 + \sigma_l^2 + \sigma_c^2} \approx 86\,\text{nm}$$

Then, we searched for tracks that contained single-particle positions, for which the two circles 'overlapped' for any of the NPC positions. To finally sort tracks for the respective categories (see Supplementary Fig. 9) we checked firstly, if the circle around the first or last detection overlapped with a circle around an NPC, then it was checked if the start or end position of a track was within the membrane polygon or outside[68,69]. At last, it was checked, if the particle positions changed unexpectedly between inside and outside of the polygon during the trajectory. Tracks were then sorted into respective categories based on their start, intermediate and end positions as indicated in Supplementary Fig. 9. The number of frames for which a particle showed an uninterrupted overlap of the circles of a track and the NPC was taken as the dwell time within the pore.

**Export track alignment.** In order to exploit the information contained in all identified export events, the respective tracks were related to the central NPC through which the export took place and rotated such that the export axis coincided with the positive $x$ axis. The extent of the rotation was based on the positions of the two directly neighbouring NPCs of the transporting NPC (Supplementary Fig. 11, steps 1–6). In the first step, all coordinates were related to the central NPC. Next, the two neighbouring NPCs were connected by a straight line (Step 2), which was assumed to be perpendicular to the transport direction (Step 3). The angle ($\phi$) between this transport direction and the $x$ axis was calculated (Step 4) and the neighbouring NPCs and the track were rotated accordingly (step 5). To consider the direction of the export correctly, the orthogonal line was extended by 10 pixels. If the endpoint of this line was within the polygon created by all NPC positions, the track and the neighbouring NPCs were rotated only by $\phi$ (Step 6.1). If the endpoint was outside the polygon, everything was rotated by $\phi + 180°$ (Step 6.2).

*Mobility analysis of pre-60S subunits using spot-on.* In 324 cells, all particle trajectories were isolated, which occurred within the nucleoplasm but not in the nucleoli. In the confocal NTF2 images, the nucleoli showed a lower fluorescence intensity compared with the rest of the nucleus. This residual fluorescence was used to roughly segment the nucleoli. First, in the transformed membrane image a threshold was set using the 'Otsu' method. These mask images were inverted and manually corrected. In the end, the area of the nucleoli was dilated 15 times. All tracks that are located within the area of this mask were excluded for further analysis.

In this way, 7785 tracks with a total number of 41,457 localisations were selected. These trajectories were analysed using the diffusion coefficient approach SpotOn[70] in order to determine the fastest diffusion coefficient. The bin width in the jump distance histograms was set 0.02 µm and the number time points to 8. Only the first four jumps were considered and the maximal jump distance was set to 1 µm (corresponding to a diffusion coefficient of 18 µm²/s). For model fitting a three-state model was chosen with the following start parameters: $D_{bound} = 0.0005–0.08$ µm²/s, $D_{slow} = 0.15–5$ µm²/s, $D_{fast} = 0.15–25$ µm²/s, $F_{bound} = F_{fast} = 0–1$, Localisation error = 0.043 µm, Model Fit = CDF, Iterations = 3.

*Kinetic data fitting.* The export dwell time kinetics were obtained by fitting an exponential decay model to the probability distribution of the export dwell time using Mathematica (version 12.1.0.0, Wolfram, Long Hanborough, UK) assuming the determined frequencies were Poisson distributed. To test if M different populations of particles existed, the dwell time distributions were modelled by single, double or triple-modal ($M \in [1–3]$) functions[71].

$$pdf\left(t_i; a, k_i, S_i, b\right) = \sum_{i=1}^{M} \left( a_i \cdot \frac{k_i^{S_i} t_i^{S_i - 1}}{\Gamma(S_i)} \cdot e^{-t_i k_i} \right) + b$$

With the following variables: $k_i$, rate constant; $S_i$, number of reaction steps; $t$, dwell time; $\Gamma$, gamma function; $a_i$, scaling factor; $b$, bias; $M$, number of modes ($M \in [1–3]$).

To select the model optimally describing the data the corrected Akaike Information Criterion ($AIC_c$) was calculated according to Akaike (1974) & Burnham et al. (2002)[39,40].

$$AIC = n\,ln\left(\frac{RSS}{n}\right) + 2k$$

$$AIC_c = AIC + \frac{2k(k+1)}{n - k - 1}$$

$n$, number of observations; RSS, residual sum of squares; $k$, number of fitted parameters.

**Determination of the sub-NPC position of NTF2.** To determine the position of NTF2 within the NPC, first HeLa S3 were transfected with HaloTag-NTF2. On the day of the experiment, cells were stained with JF549-HaloTag-Ligand and fixed with 4% paraformaldehyde in PBS for 10 min at 37 °C. Cells were washed 5× for 10 min with PBT-buffer (0.5% (v/v) TritonX in PBS with 3% (m/v) BSA) and incubated for 1 h at RT with mAb414 (MMS-120P (#902907), Covance (Biolegend), San Diego, CA, USA) at a dilution of 1:1000. Next, cells were washed 3×

with PBT-Buffer for 10 min each time. Finally, cells were incubated with a polyclonal anti-mouse secondary antibody labelled by AlexaFluor 488 (Anti-Mouse IgG (whole molecule)−Atto 488 antibody produced (goat), 62197 (Lot# BCCB9457), Sigma-Aldrich) and washed again 3× for 10 min with PBT-buffer.

The samples were analysed using a Zeiss LSM880 with an Airyscan detector. The resulting images were Airyscan processed (detector strength was set to auto) and colour shift corrected.

To determine the positions of maximum NTF2 and mAb414 accumulation, in both channels the position of the same NPC was selected manually and more precisely localised by four Gaussian fits in 45° angles. The maxima of the fits were averaged and the distance between the maxima of the NTF2- and mAb414-binding sites calculated using Excel (Microsoft, version 2007 SP3).

To find out, which maximum is on nucleoplasmic site, the position of all NPCs was determined and the centre of mass of all positions calculated. The maximum, which was closer to the centre of mass, was considered to be on the nucleoplasmic side.

**Statistics and reproducibility.** Each experiment was repeated at least three times using independent samples with similar results.

**Reporting summary.** Further information on research design is available in the Nature Research Reporting Summary linked to this article.

## Data availability
The data sets generated during the current study comprise ~500 Gigabytes of multi-channel images and were evaluated using custom-developed software routines. The raw images are available under restricted access for reasons of size and the requirement of our complex analysis strategy. Access can be obtained upon reasonable request from the corresponding author. Source data are provided with this paper.

## Material availability
All used plasmids and established cell lines are available from the corresponding author on reasonable request. Source data are provided with this paper.

## Code availability
The source code for all data analysis procedures, example image files and a PDF describing the use of the programs can be downloaded from: https://www.chemie.uni-bonn.de/pctc/kubitscheck/downloads.

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

## Acknowledgements

We gratefully acknowledge funding by the University of Bonn and a grant of the Swiss National Science Foundation (SNSF) to Ulrike Kutay (31003A_166565 and NCCR 'RNA and disease').

## Author contributions

U.Kt., J.P.S. and U.Kb. designed and coordinated the overall study. The experiments were performed by J.A.R., A.M.K., R.B. and K.D. Cell lines and constructs were produced by

R.B. and S.W., J.A.R. and D.P. developed new instrumentation. J.A.R. and A.M.K. analysed the data. J.A.R. and J.P.S. developed the code for data analysis. The manuscript was written by J.A.R., J.P.S. and U.Kb. with important contributions by R.B., K.D. and U.Kt. and further input from all authors.

## Funding

## Competing interests
The authors declare no competing interests.
