## [Peer Review File · Nature Communications]

Reviewers' Comments:

Reviewer #1:

Remarks to the Author:

In the article, 'Nuclear export of the pre-60S ribosomal subunit through single nuclear pores observed in real time', Ruland and colleagues use single-molecule live-cell microscopy to obtain mechanistic details and kinetics of 60S pre-ribosomes export in HeLa cells. To achieve this, they established a sophisticated microscope setup and developed an automated analysis pipeline allowing them to measure the behaviour of pre-60S ribosomal subunit at the nuclear periphery and during translocation through single nuclear pores in high spatial and temporal resolution. Using very stringent criteria for scoring translocation events, they could track 78 pre-60S ribosomes through single NPCs, allowing them to determine translocation kinetics and revealing a rate limiting step at the cytoplasmic side of the NPC. Moreover, the authors showed that only about one-third of export events are successful and identified a nuclear basket as the site where unsuccessful events are terminated.

Overall, this study is an impressive tour-de-force to investigate the kinetics of 60S pre-ribosome export. The experiments are of very high technical quality, are well designed and controlled, and their findings will serve as a reference in the field.

Investigating single protein behaviour in cells is challenging due to a variety of technical limitations that come with the behaviour of single molecules in cells. As described by the authors, probably the most relevant of these challenges comes from the fast diffusion of single proteins that limit the acquisition of real-time single-molecule imaging to a single imaging plane. However, as molecules diffuse in 3 dimensions, tracking molecules for more than a few tens, or sometimes, hundreds of milliseconds, is not impossible, except if these are bound to some structure that limits their diffusion, such as chromatin or a membrane. This makes studies as the one described here very challenging and is likely the main reason why the authors were only able to describe 78 export events, e.i. events where particles were able to be tracked before, during and after transport. Nevertheless, even with so few events, the authors were able to extract important kinetic information, such as translocation times, as well as a rate limiting step at the cytoplasmic side of the NPC. These observations are important to understand ribosome transport, as well as nucleocytoplasmic transport in general, and will serve as a starting point for further mechanistic studies. In my view, this *per se* merits publication. Nevertheless, the authors should address some minor issues in a revised manuscript.

Tracking single eIF6 molecules, the authors describe two nuclear eIF6 populations, one largely static in the nucleolus, and one that is nucleoplasmic with a diffusion coefficient of $1.7 \pm 0.1 \mu\text{m}^2/\text{s}$, representing eIF6 bound to pre-60S ribosomes. It is surprising that only two populations are described as at least a third population representing free eIF6 should be expected. eIF6 has to be imported to the nucleus before associating with pre-ribosomes in the nucleolus, and therefore, at least some eIF6 will be present in the cells in either an import complex or as single protein. The imaging setup and the acquisition speed of 13ms should allow tracking of fast diffusing free eIF6. The authors have to discuss why they do not observe such a population. Moreover, if they do, they should discuss the possible behaviour of such molecules with respect to the data presented. If they don't, why do they think no such population is observed?

In addition to translocations, the authors identify several different interaction events, allowing them to identify the basket as the site where unsuccessful events are terminated. Even if not required for this manuscript, it is surprising that the authors have not included data of 60S single particle behaviour at the nuclear periphery when blocking export, as they already did block export using LMB in supplemental Figure 3. With imaging setup and automated image analysis in place, such an analysis could reveal important additional mechanistic understanding of this process.

It is not evident how meaningful the description of the mass flux through a single NPC ($\sim 125 \text{MDa}\cdot\text{s}^{-1}$) is to a general audience. The author might want to better describe what this number signifies in terms of the many cargos of different size and possibly translocation times, including karyopherin-protein cargos and mRNPs.

Reviewer #2:

Remarks to the Author:

Review of Ruland et al "Nuclear export of the pre-60S ribosomal subunit through single nuclear pores observed in real time"

Nature Comm 2021

The export of ribosomes out of the nucleus is a major transport pathway in eukaryotic cells. Although the molecular players are well defined, observing ribosomal subunits transiting the nuclear pore complex has been elusive. A recent cryo-ET analysis of ribosome export in yeast from the Gleizes lab captured static views of ribosomal particles within the NPC and estimated a dwell time of about 90ms. In this manuscript, the authors use a combination of single particle and superresolution microscopy to monitor export of the large ribosomal subunit in real time and calculate a dwell time of 24 ms. The work is technically impressive, however, I have several concerns about methods and conclusions that need attention.

Major points

1) The authors use exogenously expressed eIF6-HaloTag throughout their work. A critical question is whether or not they are monitoring pre-60S export with this construct. They provide several lines of evidence in support of their conclusion. However, most of these arguments are weak or have alternative explanations. Neither the bulk nucleolar accumulation nor the loss of nucleolar fluorescence in the presence of actinomycin D are strong evidence. The co-IP of Rpl10 with eIF6 is not strong evidence either as Rpl10 assembles into pre-60S in the cytoplasm. This indicates that eIF6 associates with cytoplasmic 60S but there is evidence from work from the Beckmann and Weissman groups that Tif6 also engages recycling mature 60S subunits during RQC. Since Rpl10 joins subunits during their maturation in the cytoplasm, the presence of Rpl10 on eIF6HaloTag particles cannot be used as an evidence that the nuclear signal seen is 60S bound eIF6 and not free eIF6. The effect of LMB on eIF6 localization was extremely modest raising the possibility that the bulk of eIF6 is not sensitive to LMB and is likely not associated with pre-60S. The best evidence appears to be that the single particles the authors monitored diffused more slowly than unbound proteins. Perhaps the strongest argument is one the authors do not make, that free eIF6 should not be exported and its very presence in the NPC suggests association with pre-60S. Because this point underlies all conclusion in this manuscript, the authors are obligated to do additional experiments to support their conclusion that they are monitoring pre-60S by tracking eIF6HaloTag.

I suggest they:

a) Determine the fraction of eIF6HaloTag that is 60S associated by western blotting for eIF6 across a sucrose gradient. I am concerned that the majority of the protein will be free protein.
b) Repeat the single particle/superresolution analysis in the presence of LMB to demonstrate that eIF6HaloTag particles are not exported under these conditions. This may also reveal an initial docking step on the nuclear basket as suggested by the Gleizes group.
c) Monitor import events of eIF6HaloTag, in which the protein should be in a much smaller complex to directly compare diffusion coefficients, and show that the putative export complex has slower diffusion. These events are probably present in their existing data sets.

2) P5 "The biochemical and microscopic data were also verified ..." Verify isn't quite right. The results may be consistent. Also, more explanation is needed for the reader to understand how Actinomycin D is known to affect nucleolar structure and eIF6 localization. Citations are needed here.

3) Supplemental Fig 3. When quantifying the change in localization, the authors should compare normalized signal in cytoplasm, nucleoplasm and nucleolus, and plot each as fraction total. The change in localization is not obvious. Quantifying signal along transects may be more instructive. The Ethanol control cells look strange, even at 0min. They have altered nucleolar morphology and very little cytoplasm. A better matched control is important for this comparison because the

change in localization is quite modest.

Minor points

4. Fig 1D, please include Full Lysate for HeLaS3 to show equal loading between the two samples.

5. At the bottom of P10: In the sentence "... indicate that the translocating particles were most often observed in the region of the cytoplasmic ring of the NPC (Fig 5)." This should refer to Fig 4.

6. It is incorrect to state "Also, for the non-canonical export factors in yeast, e.g. Arx1, it is unknown how and when they are detached from the yeast pre-60S particle." It is well-established that the release of Arx1 requires Rei1 and the along with the ATPase Ssa and its cofactor Jjj1. This should be corrected.

7. The authors write ". Our finding of a single rate limiting step and single maximum of binding sites in the central NPC suggests a model, where both Exp5 and Crm1 are needed for NPC entry and passage of the pre-60S particle, but only the Nup214/Crm1 interaction is needed for release from the NPC. Here, the strong and direct interaction between Nup214/Crm1 would 'fish-out' the pre-60S export complex from the central NPC channel via FG-repeats50 and hand it over to NUP358, which results in GTP-hydrolysis and subsequent cytosolic pre-60S release. " However, this understanding has already been established by the previous contributions of multiple labs.

8. Page 16 " Thus, only about 35% of attempted export events were successful. Similar fractions have been reported for mRNPs", please provide reference here.

9. In several places "events" should be used instead of "processes:"

"Altogether, we identified 78 complete export processes"

"the category "cytoplasmic dissociation" (Fig. 3G and Table 1) contained processes"

"Obviously, also the category "nucleoplasmic dissociation" comprising 229 processes"

"Interestingly, we observed this significantly more often (n=145) than successful export processes"

Reviewer #3:

Remarks to the Author:

Ruland et al. investigate the transport kinetics of the large ribosomal pre-60S particle through single NPCs in living cells. A fair number of studies have explored how such export complexes assemble in the nucleolus. However, little is known about the transport kinetics of pre-60S at the single NPC and single particle level.

To address this central biological question, the authors track in real-time the NPC transport of eIF6-HaloTag labelled pre-60S particles in HeLa cells. They deduce the dwell times and percentage of the successful events by combining Airyscan super resolution laser scanning confocal microscopy and single particle tracking. Moreover, different interaction modes were categorised, which show that transient interactions actually dominate pre-60S export complexes at the NPC periphery.

The microscopy data is of a high quality and is statistically significant. However, it is generally challenging to know for sure how algorithms treat complex data sets - and this is no exception - although credit has to be given to the authors for going the extra mile to explain them.

Questions/comments to be addressed:

1. EGFP-NTF2 was used as an internal reference within the NPC to correlate the RNP trajectories. However, as the authors themselves had previously shown (Kubitscheck 2005), NTF2, which delivers RanGDP into the cell nucleus, itself binds transiently to the NPC with a dwell time of 5-6 ms. Other studies have also shown that NTF2 binds more weakly to the FG Nups, such as in comparison to importinB (Wagner 2015).

Q: Can the authors please comment on how stable this pool of EGFP-NTF2 is at the NPC? Does the EGFP signal fluctuate (in position and intensity) over time, in comparison to labelling a membrane or scaffold Nup such as POM121 (Grünwald 2011)? What are the levels of EGFP-NTF2 in comparison to endogenous NTF2? Does EGFP-NTF2 still transport RanGDP?

2. The EGFP-NTF2 fluorescence intensity was used "as an indicator for the NPC middle location". This peak is at ca. -24 nm from the NPC middle plane position i.e., towards the nuclear basket, based on Mab414 antibody staining, which is known to recognize Nup62. However, it was previously reported that Nup62 is localized at ca. +10 nm from the midplane towards the cytoplasmic side. Hence, the distance between the peak for eGFP-NTF2 here and peak for the Mab414 is ca. 34 nm.

Q: Still, it is not clear how the authors defined the position of Nup62 in the NPC using Mab414. From Fig. 2a/b in Schwarz-Herion 2007, it is the C-terminus of Nup62 that is localized at +10 nm from the central plane whereas the N-terminus has more of a bimodal distribution at ca. +17 and -20 nm (Fig. 2c/d). Yet, Mab414 should only recognize the FG domain of Nup62, which is located at the N-terminus. Can the authors please clarify?

Q: Mab414 ought to interact with other FG Nups besides Nup62. How do the authors know with certainty that only Nup62 was targeted?

3. Pg 15 and Fig. 5. "Our measured distribution of trajectory positions, although less well resolved than the EM data, showed a clear, single maximum in the central region of the NPC shifted towards the cytoplasm."

Q: However, in Fig. 5B the maximum seems to be clearly in the region of the nuclear basket. Can the authors please comment on this or clarify?

4. The authors state "Ribbeck and Görlich (2002) found that the transport rates of bigger transport cargoes scale with the number of loaded transport receptors" then speculate that each pre-60S particle is accompanied by one copy of CRM1 and more than one copy of Exp5.

Q: It may be good if the authors can expand their discussion on this point. Could the observed modes of behaviour (Fig. 3G) be influenced by the number of bound NTRs, especially Exp5? See also Paci eLIFE 2020 and Tu EMBOJ 2013.

5. What is the difference between mean translocation time, export duration, and dwell time?

6. Refs 52 and 55 are replicates.

The technology on display is impressive, and the scientific insights are novel, significant and timely. Hence, I would support its publication in Nature Communications once the above questions are clarified.

We would like to thank the reviewers for their positive evaluation, their careful reading and detailed review of our manuscript that yielded a substantial further improvement. We have addressed the various questions and suggestions as they were raised in the assessments.

Reviewer #1 (Remarks to the Author):

In the article, 'Nuclear export of the pre-60S ribosomal subunit through single nuclear pores observed in real time', Ruland and colleagues use single-molecule live-cell microscopy to obtain mechanistic details and kinetics of 60S pre-ribosomes export in HeLa cells. To achieve this, they established a sophisticated microscope setup and developed an automated analysis pipeline allowing them to measure the behaviour of pre-60S ribosomal subunit at the nuclear periphery and during translocation through single nuclear pores in high spatial and temporal resolution. Using very stringent criteria for scoring translocation events, they could track 78 pre-60S ribosomes through single NPCs, allowing them to determine translocation kinetics and revealing a rate limiting step at the cytoplasmic side of the NPC. Moreover, the authors showed that only about one-third of export events are successful and identified a nuclear basket as the site where unsuccessful events are terminated.

Overall, this study is an impressive tour-de-force to investigate the kinetics of 60S pre-ribosome export. The experiments are of very high technical quality, are well designed and controlled, and their findings will serve as a reference in the field.

We would like to thank the referee for taking the time to review the paper, and for the positive assessment of our work.

Investigating single protein behaviour in cells is challenging due to a variety of technical limitations that come with the behaviour of single molecules in cells. As described by the authors, probably the most relevant of these challenges comes from the fast diffusion of single proteins that limit the acquisition of real-time single-molecule imaging to a single imaging plane. However, as molecules diffuse in 3 dimensions, tracking molecules for more than a few tens, or sometimes, hundreds of milliseconds, is not impossible, except if these are bound to some structure that limits their diffusion, such as chromatin or a membrane. This makes studies as the one described here very challenging and is likely the main reason why the authors were only able to describe 78 export events, e.i. events where particles were able to be tracked before, during and after transport. Nevertheless, even with so few events, the authors were able to extract important kinetic information, such as translocation times, as well as a rate limiting step at the cytoplasmic side of the NPC. These observations are important to understand ribosome transport, as well as nucleocytoplasmic transport in general, and will serve as a starting point for further mechanistic studies. In my view, this per se merits publication. Nevertheless, the authors should address some minor issues in a revised manuscript.

We completely agree with the assumption of the reviewer with regard to the origin of the low number of export event observations. It was our priority to have a solid data base, even if it might be small.

Tracking single eIF6 molecules, the authors describe two nuclear eIF6 populations, one largely static in the nucleolus, and one that is nucleoplasmic with a diffusion coefficient of $1.7 \pm 0.1 \mu\text{m}^2/\text{s}$, representing eIF6 bound to pre-60S ribosomes. It is surprising that only two populations are described as at least a third population representing free eIF6 should be expected. eIF6 has to be imported to the nucleus before associating with pre-ribosomes in the nucleolus, and therefore, at least some eIF6 will be present in the cells in either an import complex or as single protein. The imaging setup and the acquisition speed of 13ms should allow tracking of fast diffusing free eIF6. The authors have to discuss why they do not observe such a population. Moreover, if they do, they should discuss the possible behaviour of such molecules with respect to the data presented. If they don't, why do they think no such population is observed?

The reviewer is completely correct. The diffusion coefficient of free eIF6-HaloTag can be estimated to be $20 \mu\text{m}^2/\text{s}$. That means that they will cover a root mean square distance of $1.05 \mu\text{m}$ within 14 ms (the inverse of the frame rate). When we plot the jump distance distribution of the analyzed trajectories together with the fit result, it can be noted that there exists a fraction of very fast diffusing particles:

However, their fraction was quite small (because it can be assumed that most fast particles leave the focal plane) and a quantification was problematic. To address this topic, we added the following sentence on page 6 to the manuscript: “The corresponding jump distance distribution indicated also a small fraction of significantly faster diffusing particles. Presumably, these jumps were due to free eIF6-HaloTag-JF549 molecules.”

In addition to translocations, the authors identify several different interaction events, allowing them to identify the basket as the site where unsuccessful events are terminated. Even if not required for this manuscript, it is surprising that the authors have not included data of 60S single particle behaviour at the nuclear periphery when blocking export, as they already did block export using LMB in supplemental Figure 3. With imaging setup and automated image analysis in place, such an analysis could reveal important additional mechanistic understanding of this process.

We agree with the reviewer that the respective data would improve the manuscript. We performed new single particle tracking experiments and detected 75 “failed export events” in 74 movies. Thus, we added now the distribution of observation positions according to Fig. 5B as a new supplemental Fig. 11B (120 min LMB), where we compared it to the distribution without LMB (Fig. 5B). Notably, the maximum of both

distributions was located at the nuclear basket.

It is not evident how meaningful the description of the mass flux through a single NPC ($\sim 125 \text{ MDa}\cdot\text{s}^{-1}$) is to a general audience. The author might want to better describe what this number signifies in terms of the many cargos of different size and possibly translocation times, including karyopherin-protein cargos and mRNPs.

Principally we do agree. However, we did not want to elaborate too much on this point, this might rather be done in a future discussion article. We modified the respective paragraph as follows:

This value is in very good agreement with estimations for the maximum transport rate of cargo molecules through the human NPC formerly estimated by Ribbeck and Görlich (2001, 2002)^{35,47}. Based on their bulk measurements they assumed that ~ 1.000 importin β per second can translocated through a single NPC, which equates to a mass flux of roughly $100 \text{ MDa}\cdot\text{s}^{-1}$. From the electron tomography study of Delavoie et al (2019) it is evident that ribosomal subunits are exported one at a time and not in parallel. The translocation duration of 24 ± 4 ms in principle would allow the transport of ~ 35 to 50 large subunits per second through a single pore. Assuming an average molecular mass of $\sim 2.5 \text{ MDa}$ for the human pre-60S particle this accounts for a mass flux of 87.5 to $125 \text{ MDa}\cdot\text{s}^{-1}$ per NPC or ~ 875 - 1250 importin β s^{-1} per NPC.

Reviewer #2 (Remarks to the Author):

The export of ribosomes out of the nucleus is a major transport pathway in eukaryotic cells. Although the molecular players are well defined, observing ribosomal subunits transiting the nuclear pore complex has been elusive. A recent cryo-ET analysis of ribosome export in yeast from the Gleizes lab captured static views of ribosomal particles within the NPC and estimated a dwell time of about 90ms. In this manuscript, the authors use a combination of single particle and superresolution microscopy to monitor export of the large ribosomal subunit in real time and calculate a dwell time of 24 ms. The work is technically impressive, however, I have several concerns about methods and conclusions that need attention.

We would like to thank the referee for taking the time to review the paper, and for appreciating our work.

Major points

1) The authors use exogenously expressed eIF6-HaloTag throughout their work. A critical question is whether or not they are monitoring pre-60S export with this construct. They provide several lines of evidence in support of their conclusion. However, most of these arguments are weak or have alternative explanations. Neither the bulk nucleolar accumulation nor the loss of nucleolar fluorescence in the presence of actinomycin D are strong evidence. The co-IP of Rpl10 with eIF6 is not strong evidence either as Rpl10 assembles into pre-60S in the cytoplasm. This indicates that eIF6 associates with cytoplasmic 60S but there is evidence from work

from the Beckmann and Weissman groups that Tif6 also engages recycling mature 60S subunits during RQC. Since Rpl10 joins subunits during their maturation in the cytoplasm, the presence of Rpl10 on eIF6HaloTag particles cannot be used as an evidence that the nuclear signal seen is 60S bound eIF6 and not free eIF6. The effect of LMB on eIF6 localization was extremely modest raising the possibility that the bulk of eIF6 is not sensitive to LMB and is likely not associated with pre-60S. The best evidence appears to be that the single particles the authors monitored diffused more slowly than unbound proteins. Perhaps the strongest argument is one the authors do not make, that free eIF6 should not be exported and its very presence in the NPC suggests association with pre-60S. Because this point underlies all conclusion in this manuscript, the authors are obligated to do additional experiments to support their conclusion that they are monitoring pre-60S by tracking eIF6HaloTag.

We thank the reviewer for raising this point, which is critical for the validity of our study. In this context we would like to stress our result that the mobility of the observed particles was indeed not only lower than that of free eIF6-HaloTag (see also Top 1 of Reviewer 1), but also lower than that of pre-40S particles.

I suggest they:

a) Determine the fraction of eIF6HaloTag that is 60S associated by western blotting for eIF6 across a sucrose gradient. I am concerned that the majority of the protein will be free protein.

We followed the suggestion of the reviewer and performed the suggested Western blot. As already indicated by the LMB blocking experiment and suspected here by the reviewer (and us, as we stated in the manuscript on page 5), the result was that the amount of unbound eIF6-HaloTag is substantial:

Legend: Extract of HeLa cells expressing EIF6-HaloTag cells was separated on a 10-45% sucrose gradient by centrifugation. Input and gradient fractions were analyzed by immunoblotting using the indicated antibodies.

In order to account for this topic, we added this Western blot to the supplemental material (new supplemental Fig. 3D) and modified the last sentence on page 5: "This view was also supported by a determination of the fraction of eIF6-HaloTag that was pre-60S particle-associated by western blotting for eIF6 across a sucrose gradient (supplemental Figure 3D)."

b) Repeat the single particle/superresolution analysis in the presence of LMB to demonstrate that eIF6HaloTag particles are not exported under these conditions. This may also reveal an initial docking step on the nuclear basket as suggested by the Gleizes group.

We performed these experiments as suggested. To this end, we analyzed movies from 81 different cells with an integrated run time of 26,7 min (50 min after addition of

LMB) and from 42 different cells with an integrated run time of 15,5 min 120 min after addition of LMB). As expected, the presence of LMB lead to a dramatic reduction in the observation of export events from 0.92 events/min to 0.13 events/min:

Number of detected pre-60S export events in the absence, and 50 and 120 after addition of LMB, respectively.

We added the following remark to the manuscript on page 12:

“Interaction of single pre-60S particles with NPCs in the presence of LMB. Finally, we examined the nuclear export of pre-60S particles upon addition of LMB, which inhibits export as discussed above. As expected, the observation of export events was reduced from 0.91 events/min to 0.13 events/min 120 min after addition of LMB and thus became very rare (supplemental Fig. 11A). This inhibition confirmed that the observed nuclear export events (Figs. 3 and 4) were due to eIF6-HaloTag bound to pre-60S particles.” Also, we added the figure to the supplemental material (supplemental Fig. 11A). As stated in the response to Reviewer 1 the maximum position of the failed export event distribution, however, did not change when comparing it to the situation without LMB.

c) Monitor import events of eIF6HaloTag, in which the protein should be in a much smaller complex to directly compare diffusion coefficients, and show that the putative export complex has slower diffusion. These events are probably present in their existing data sets.

We observed in all our experiments only two (!) bona fide import events (Table 1) – not enough for a meaningful discussion of diffusion coefficients. As stated in the text and outlined in the answer to reviewer 1, the frame rate of 71.4 frames/s is not high enough to unambiguously follow the diffusive motion of single proteins.

2) P5 “The biochemical and microscopic data were also verified ...” Verify isn’t quite right. The results may be consistent. Also, more explanation is needed for the reader to understand how Actinomycin D is known to affect nucleolar structure and eIF6 localization. Citations are needed here.

Amended. We improved the respective paragraph on page 5 as follows: “The biochemical and microscopic data were also consistent with the result of an incubation of the eIF6-JF549 cells with the transcriptional inhibitor Actinomycin D. RNA polymerase I catalysing ribosomal RNA transcription is most sensitive to the drug, which abolishes ribosome biogenesis and causes a redistribution of nucleolar components including nascent pre-ribosomes. We observed a complete loss of nucleolar labelling suggesting an incorporation of eIF6-HaloTag into pre-60S particles (supplemental Figure 2).” In addition, two new references were given.

3) Supplemental Fig 3. When quantifying the change in localization, the authors

should compare normalized signal in cytoplasm, nucleoplasm and nucleolus, and plot each as fraction total. The change in localization is not obvious. Quantifying signal along transects may be more instructive. The Ethanol control cells look strange, even at 0min. They have altered nucleolar morphology and very little cytoplasm. A better matched control is important for this comparison because the change in localization is quite modest.

We agree to the last point. The Ethanol control was repeated, and the image was replaced. We tried various options for presenting the quantitative analysis of the fluorescence intensities in cytoplasm, nucleoplasm and nucleoli. However, we did not feel that a different representation would reflect the change in localization in a better way.

Minor points

4. Fig 1D, please include Full Lysate for HeLaS3 to show equal loading between the two samples.

The figure was accordingly amended.

5. At the bottom of P10: In the sentence "... indicate that the translocating particles were most often observed in the region of the cytoplasmic ring of the NPC (Fig 5)." This should refer to Fig 4.

Amended.

6. It is incorrect to state "Also, for the non-canonical export factors in yeast, e.g. Arx1, it is unknown how and when they are detached from the yeast pre-60S particle." It is well-established that the release of Arx1 requires Rei1 and the along with the ATPase Ssa and its cofactor Jjj1. This should be corrected.

We agree to the reviewer. We deleted this wrong statement.

7. The authors write ". Our finding of a single rate limiting step and single maximum of binding sites in the central NPC suggests a model, where both Exp5 and Crm1 are needed for NPC entry and passage of the pre-60S particle, but only the Nup214/Crm1 interaction is needed for release from the NPC. Here, the strong and direct interaction between Nup214/Crm1 would 'fish-out' the pre-60S export complex from the central NPC channel via FG-repeats50 and hand it over to NUP358, which results in GTP-hydrolysis and subsequent cytosolic pre-60S release. " However, this understanding has already been established by the previous contributions of multiple labs.

Amended: we replaced "suggests" by "confirms" and added further references to this model. It was not our intention to claim that we developed it.

8. Page 16 " Thus, only about 35% of attempted export events were successful. Similar fractions have been reported for mRNPs", please provide reference here.

Amended.

9. In several places "events" should be used instead of "processes:"

“Altogether, we identified 78 complete export processes”
“the category “cytoplasmic dissociation“ (Fig. 3G and Table 1) contained processes”
“Obviously, also the category “nucleoplasmic dissociation” comprising 229 processes”
“Interestingly, we observed this significantly more often (n=145) than successful export processes”

Amended.

Reviewer #3 (Remarks to the Author):

Ruland et al. investigate the transport kinetics of the large ribosomal pre-60S particle through single NPCs in living cells. A fair number of studies have explored how such export complexes assemble in the nucleolus. However, little is known about the transport kinetics of pre-60S at the single NPC and single particle level.

To address this central biological question, the authors track in real-time the NPC transport of eIF6-HaloTag labelled pre-60S particles in HeLa cells. They deduce the dwell times and percentage of the successful events by combining Airyscan super resolution laser scanning confocal microscopy and single particle tracking. Moreover, different interaction modes were categorised, which show that transient interactions actually dominate pre-60S export complexes at the NPC periphery.

The microscopy data is of a high quality and is statistically significant. However, it is generally challenging to know for sure how algorithms treat complex data sets - and this is no exception – although credit has to be given to the authors for going the extra mile to explain them.

We would like to thank the referee for taking the time to review the paper, and for appreciating our efforts.

Questions/comments to be addressed:

1. EGFP-NTF2 was used as an internal reference within the NPC to correlate the RNP trajectories. However, as the authors themselves had previously shown (Kubitscheck 2005), NTF2, which delivers RanGDP into the cell nucleus, itself binds transiently to the NPC with a dwell time of 5-6 ms. Other studies have also shown that NTF2 binds more weakly to the FG Nups, such as in comparison to importinB (Wagner 2015).

Q: Can the authors please comment on how stable this pool of EGFP-NTF2 is at the NPC? Does the EGFP signal fluctuate (in position and intensity) over time, in comparison to labelling a membrane or scaffold Nup such as POM121 (Grünwald 2011)? What are the levels of EGFP-NTF2 in comparison to endogenous NTF2? Does EGFP-NTF2 still transport RanGDP?

The reviewer raises an important topic. For observing single particle transport across the NPC a fluorescence marker for the latter is needed. In several studies by various labs – e.g. the Musser, Yang, Grünwald, Zenklusen, Shav-Tal and our lab – autofluorescent proteins were used to this end. Fusion proteins of structural nucleoporins with auto-fluorescent proteins provide a firmly attached, however

relative weak fluorescent label in terms of fluorophore number and thus intensity, which is also prone to bleaching. In 2012 we used successfully recombinant NTF2, which was fluorescence labelled and co-microinjected to identify NPCs with a transient but replenishable fluorescence marker (Siebrasse et al., 2012). In this new study here, we used therefore stably co-expressed GFP-tagged hNTF2. We now show in the new Supplemental Figure 4 that this approach allows a longer and more precise observation of the NPC than using stably NPC-incorporated fusion proteins, e.g. POM121-GFP.

To specifically address the various questions:

Stability of the EGFP-NTF2 pool at the NPC

The binding of EGFP-NTF2 to the NPC in permeabilized cells is short-lived with a binding time of 5.8 ms (Kubitscheck et al., 2005), but probably more complex in molecularly crowded surroundings (Wagner et al., 2015). To address the situation in living cells we measured the dynamics of eGFP-NTF2 at the nuclear envelope using FRAP. We observed a rapid recovery of the bleached eGFP-NTF2 at the nuclear envelope with an estimated half-time of 0.2 to 0.4 ms. The mobile fraction was roughly 90%. This demonstrated a rapid exchange of the fluorescent label and should result in an prolonged localization precision of NPC (see below).

Fluctuation of eGFP-NTF2 in intensity over time in comparison to POM121-eGFP

To address this we measured the fluorescence intensity of single pores as a function of time using our standard illumination conditions.

Mean fluorescence intensity of single NPCs as a function of time. Airyscan images of HeLa S3 cells stably expressing eGFP-NTF2 or eGFP-Pom121 were acquired as described in Methods. One scan lasted 1.74 s. These images were acquired under the same conditions with regard to the irradiance per scan as the images that were used to determine the NPC positions in the paper. The resulting membrane images were aligned using the StackReg plug-in for ImageJ. Pore positions were determined (see Methods) and the intensity of all individual pores was determined as a function of time. For NTF2, the fluorescence intensity was normalized and averaged over 306 individual NPCs from 5 different cells, and for Pom121, over 332 NPCs from 7 cells.

Thus, the average intensity is higher, and therefore also the relative intensity fluctuations will be lower. Importantly this experiment revealed that – as anticipated – the labeling of NPCs by EGFP-NTF2 photobleached less rapidly than the labeling by POM121-GFP, probably due to the replenishment of the protein.

Fluctuation of EGFP-NTF2 in position over time in comparison to POM121-GFP

To address this we determined the position of single NPCs as a function of time under the used illumination conditions for the single particle tracking experiments and plotted their mean square displacement related to the initial position. Thus, we observed a positional stability of ≤ 1 nm for a time span of < 20 s, which corresponded to our total measurement duration. For comparison, Grünwald & Singer (2010) reported a value of ± 15 nm for the localization precision of POM121 fused to tandem Tomato. They stated, however, that their NPCs were labeled only with one to three copies of tandem Tomato.

What are the levels of EGFP-NTF2 in comparison to endogenous NTF2?

To answer this question we now included Western Blots of the cell lysate using antibodies against hNTF2 and GFP (supplemental Figure 4A). GFP-tagged NTF2 is expressed at a similar level like the endogenous NTF2.

Does EGFP-NTF2 still transport RanGDP?

This cannot be answered directly from our data. However, like the wild-type NTF2 the eGFP-fusion protein also displayed a strong and specific NPC enrichment. This FG interaction of NTF2 requires the homo-dimerisation of the protein (Bayliss et al., 2002). If dimerisation is abolished, e.g. through mutation of certain residues at the dimer interface (Bayliss et al., 2002) or excess labelling of the NTF2 protein (Siebrasse et al., 2012), the resulting NTF2 monomer is no longer accumulated at the NPC. We suspect therefore that eGFP-NTF2 still transports RanGDP, since homo-dimerisation is also a prerequisite for RanGDP transport, but we have no direct evidence. For our study it is important that the eGFP-tagged NTF2 forms functional homo-dimers and recognizes the FG-repeats like the wild-type protein.

2. The EGFP-NTF2 fluorescence intensity was used “as an indicator for the NPC middle location”. This peak is at ca. -24 nm from the NPC middle plane position i.e., towards the nuclear basket, based on Mab414 antibody staining, which is known to recognize Nup62. However, it was previously reported that Nup62 is localized at ca. +10 nm from the midplane towards the cytoplasmic side. Hence, the distance between the peak for eGFP-NTF2 here and peak for the Mab414 is ca. 34 nm.

Q: Still, it is not clear how the authors defined the position of Nup62 in the NPC using Mab414. From Fig. 2a/b in Schwarz-Herion 2007, it is the C-terminus of Nup62 that is localized at +10 nm from the central plane whereas the N-terminus has more of a bimodal distribution at ca. +17 and -20 nm (Fig. 2c/d). Yet, Mab414 should only recognize the FG domain of Nup62, which is located at the N-terminus. Can the authors please clarify?

We thank the reviewer for raising this point. Indeed, the precise localization of p62 – especially by mAb414 – is not straightforward. Therefore, we reconsidered the literature and repeated our calibration based on an EM analysis of mAb414 itself without referring to Nup62, namely the publication by Grote et al., 1995 (M. Grote, U. Kubitscheck, R. Reichelt and R. Peters. Mapping of nucleoporins to the center of the nuclear pore complex by post-embedding immunogold electron microscopy. *Journal of Cell Science* 108, 2963–2972, 1995). In this publication the localization of binding sites specifically of mAb414 (and an anti-p62 antibody) was examined in 3T3 cells, human keratinocytes and rat liver cells. One result was that the binding site distributions were comparable. In their figure 6 they show the binding of mAb414 to

NPCs in rat liver cells. We re-evaluated these data with regard to the position of the maximum of the binding site distribution and yielded the following result:

Binding site distribution of mAb414 along the axis of the NPC ($z=0$ corresponds to the central plane of the NPC, negative numbers refer to nucleoplasmic positions) according to Fig. 6g and 6h of Grote et al., 2021.

The mean value of this distribution was 5 ± 25 nm, the median was 4 nm. We would like to stress that these values differed by the mean position of mAb414 used in our work so far only by 5 nm and thus the difference does not affect any of our conclusions. Nevertheless, we used the value of 5 nm for our data analysis, modified Figs. 4B and 5B accordingly and reformulated the respective sections in the manuscript as follows:

In the manuscript on page 10:

The location of the maximum NTF2 fluorescence was related to the NPC topology using mAb414 labelling Nup 62 and various other the nucleoporins (Davis and Blobel, 1986) as reference (supplemental Fig. 9 and Methods). Thereby, we found that NTF2 binds preferentially between 0 and 70 nm off the central NPC plane with its maximum at 29 nm towards the nuclear basket.

In the Methods on page 11:

To determine the positions of maximum NTF2 and mAb414 accumulation, in both channels the position of the same NPC was selected manually and more precisely localized by four Gaussian fits in 45° angles. The maxima of the fits were averaged and the distance between the maxima of the NTF2 and mAb414 calculated.

We also changed the legend to supplemental Fig. 9 accordingly.

Q: Mab414 ought to interact with other FG Nups besides Nup62. How do the authors know with certainty that only Nup62 was targeted?

The reviewer is correct that mAb414 does not only bind Nup62. Therefore, we now used EM-based data of the binding of mAb414 (Grote et al., 1995) as a reference for determining the NTF2 position as outlined in the answer to the previous question.

3. Pg 15 and Fig. 5. “Our measured distribution of trajectory positions, although less well resolved than the EM data, showed a clear, single maximum in the central region of the NPC shifted towards the cytoplasm.”

Q: However, in Fig. 5B the maximum seems to be clearly in the region of the nuclear basket. Can the authors please comment on this or clarify?

We are very sorry. The reference on page 15 given in this context must be to Fig. 4B and not to Fig. 5. We corrected this. Fig. 4B shows the mentioned result.

4. The authors state “Ribbeck and Görlich (2002) found that the transport rates of bigger transport cargoes scale with the number of loaded transport receptors” then speculate that each pre-60S particle is accompanied by one copy of CRM1 and more than one copy of Exp5.

Q: It may be good if the authors can expand their discussion on this point. Could the observed modes of behaviour (Fig. 3G) be influenced by the number of bound NTRs, especially Exp5? See also Paci eLIFE 2020 and Tu EMBOJ 2013.

We followed this suggestion of the reviewer and now wrote on page 14: “Ribbeck and Görlich (2002) found that the transport rates of bigger transport cargoes scale with the number of loaded transport receptors⁴⁶. This was later confirmed by studies of nuclear import of exceptionally big artificial transport substrates in permeabilized mammalian cells⁴⁷⁻⁴⁹. Lowe et al. (2010) used quantum dots (QD) functionalized with importin β binding domains featuring a hydrodynamic radius of ~ 18 nm⁴⁷. Reducing the number of bound importin β molecules dramatically increased the QD dwell time in the central NPC channel⁴⁷. Tu et al. (2013) engineered β -galactosidase (β -gal) to carry four M9-signals resulting in an import cargo of comparable size ($18 \times 15 \times 9$ nm³). Their single molecule analysis revealed that loading the β -gal with one receptor allowed NPC binding but no translocation. For maximum import efficiency loading of four transportins per β -gal was necessary⁴⁸. In a more recent study⁴⁹ the import of artificial virus capsids with increasing diameters (17 to 36 nm) and varying numbers of NLS (zero to 240) was quantified in ensemble measurements. Again, the initial transport rate and efficiency scaled with the number of NLS/transport receptors attached to the capsids. Together, these studies illustrate that the enthalpic and entropic costs of “solving” a large cargo in the dense FG network can be compensated by multiple NTR-FG interactions.”

5. What is the difference between mean translocation time, export duration, and dwell time?

Mean translocation time corresponds to the mean value of all dwell times. Dwell time corresponds to the interaction duration of single exported particles according to our definition given on page 8. Expectation value for the export duration: time constant determined by fitting an exponential decay function to the data of Fig. 4A indicated by the blue line in this figure. In the legend to Fig. 4A we wrote erroneously “export durations” where it should read “dwell times”. This was corrected. Probably this mistake contributed to a misunderstanding of the various designations.

6. Refs 52 and 55 are replicates.

Corrected.

The technology on display is impressive, and the scientific insights are novel, significant and timely. Hence, I would support its publication in Nature Communications once the above questions are clarified.

We thank the reviewer for this positive assessment of our work and hope that we addressed all concerns adequately.

Reviewers' Comments:

Reviewer #1:

Remarks to the Author:

In this revised manuscript, the authors have addressed most of the reviewers' concerns by performing additional control experiments and adding clarifications in the text. This has further improved the quality of the manuscript and I now consider this manuscript suited for publication in nature communication in its current form.

Reviewer #2:

Remarks to the Author:

The authors have addressed my concerns. This is a nice study and merits publication.

I thank the authors for their clarifications. My comments are marked in blue.

Reviewer #3 (Remarks to the Author):

Ruland et al. investigate the transport kinetics of the large ribosomal pre-60S particle through single NPCs in living cells. A fair number of studies have explored how such export complexes assemble in the nucleolus. However, little is known about the transport kinetics of pre-60S at the single NPC and single particle level.

To address this central biological question, the authors track in real-time the NPC transport of eIF6-HaloTag labelled pre-60S particles in HeLa cells. They deduce the dwell times and percentage of the successful events by combining Airyscan super resolution laser scanning confocal microscopy and single particle tracking. Moreover, different interaction modes were categorised, which show that transient interactions actually dominate pre-60S export complexes at the NPC periphery.

The microscopy data is of a high quality and is statistically significant. However, it is generally challenging to know for sure how algorithms treat complex data sets - and this is no exception – although credit has to be given to the authors for going the extra mile to explain them.

We would like to thank the referee for taking the time to review the paper, and for appreciating our efforts.

Questions/comments to be addressed:

1. EGFP-NTF2 was used as an internal reference within the NPC to correlate the RNP trajectories. However, as the authors themselves had previously shown (Kubitscheck 2005), NTF2, which delivers RanGDP into the cell nucleus, itself binds transiently to the NPC with a dwell time of 5-6 ms. Other studies have also shown that NTF2 binds more weakly to the FG Nups, such as in comparison to importinB (Wagner 2015).

Q: Can the authors please comment on how stable this pool of EGFP-NTF2 is at the NPC? Does the EGFP signal fluctuate (in position and intensity) over time, in comparison to labelling a membrane or scaffold Nup such as POM121 (Grünwald 2011)? What are the levels of EGFP-NTF2 in comparison to endogenous NTF2? Does EGFP-NTF2 still transport RanGDP?

My concerns are mostly clarified, with just a few minor points (see appended):

The reviewer raises an important topic. For observing single particle transport across the NPC a fluorescence marker for the latter is needed. In several studies by various labs – e.g. the Musser, Yang, Grünwald, Zenklusen, Shav-Tal and our lab – autofluorescent proteins were used to this end. Fusion proteins of structural nucleoporins with auto-fluorescent proteins provide a firmly attached, however relative weak fluorescent label in terms of fluorophore number and thus intensity, which is also prone to bleaching. In 2012 we used successfully recombinant NTF2, which was fluorescence labelled and co-microinjected to identify NPCs with a

transient but replenishable fluorescence marker (Siebrasse et al., 2012). In this new study here, we used therefore stably co-expressed GFP-tagged hNTF2. We now show in the new Supplemental Figure 4 that this approach allows a longer and more precise observation of the NPC than using stably NPC-incorporated fusion proteins, e.g. POM121-GFP.

To specifically address the various questions:

Stability of the EGFP-NTF2 pool at the NPC

The binding of EGFP-NTF2 to the NPC in permeabilized cells is short-lived with a binding time of 5.8 ms (Kubitscheck et al., 2005), but probably more complex in molecularly crowded surroundings (Wagner et al., 2015). To address the situation in living cells we measured the dynamics of eGFP-NTF2 at the nuclear envelope using FRAP. We observed a rapid recovery of the bleached eGFP-NTF2 at the nuclear envelope with an estimated half-time of **0.2 to 0.4 ms**. The mobile fraction was roughly 90%. This demonstrated a rapid exchange of the fluorescent label and should result in an prolonged localization precision of NPC (see below).

Typo: In Supp Fig. 4B, it is written "The estimated recovery time is 0.2 to 0.4 s".

Fluctuation of eGFP-NTF2 in intensity over time in comparison to POM121-eGFP

To address this we measured the fluorescence intensity of single pores as a function of time using our standard illumination conditions.

Mean fluorescence intensity of single NPCs as a function of time. Airyscan images of HeLa S3 cells stably expressing eGFP-NTF2 or eGFP-Pom121 were acquired as described in Methods. One scan lasted 1.74 s. These images were acquired under the same conditions with regard to the irradiance per scan as the images that were used to determine the NPC positions in the paper. The resulting membrane images were aligned using the StackReg plug-in for ImageJ. Pore positions were determined (see Methods) and the intensity of all individual pores was determined as a function of time. For NTF2, the fluorescence intensity was normalized and averaged over 306 individual NPCs from 5 different cells, and for Pom121, over 332 NPCs from 7 cells.

Thus, the average intensity is higher, and therefore also the relative intensity fluctuations will be lower. Importantly this experiment revealed that – as anticipated – the labeling of NPCs by EGFP-NTF2 photobleached less rapidly than the labeling by POM121-GFP, probably due to the replenishment of the protein.

OK

Fluctuation of EGFP-NTF2 in position over time in comparison to POM121-GFP

To address this we determined the position of single NPCs as a function of time under the used illumination conditions for the single particle tracking experiments and plotted their mean square displacement related to the initial position. Thus, we observed a positional stability of ≤ 1 nm for a time span of < 20 s, which corresponded to our total measurement duration. For comparison, Grünwald & Singer (2010) reported a value of ± 15 nm for the localization precision of POM121 fused to tandem Tomato. They stated, however, that their NPCs were labeled only with one to three copies of tandem Tomato.

OK

What are the levels of EGFP-NTF2 in comparison to endogenous NTF2?

To answer this question we now included Western Blots of the cell lysate using antibodies against hNTF2 and GFP (supplemental Figure 4A). GFP-tagged NTF2 is expressed at a similar level like the endogenous NTF2.

OK. However, a loading control is missing in Supp. Fig. 4A.

Does EGFP-NTF2 still transport RanGDP?

This cannot be answered directly from our data. However, like the wild-type NTF2 the eGFP-fusion protein also displayed a strong and specific NPC enrichment. This FG interaction of NTF2 requires the homo-dimerisation of the protein (Bayliss et al., 2002). If dimerisation is abolished, e.g. through mutation of certain residues at the dimer interface (Bayliss et al., 2002) or excess labelling of the NTF2 protein (Siebrasse et al., 2012), the resulting NTF2 monomer is no longer accumulated at the NPC. We suspect therefore that eGFP-NTF2 still transports RanGDP, since homo-dimerisation is also a prerequisite for RanGDP transport, but we have no direct evidence. For our study it is important that the eGFP-tagged NTF2 forms functional homo-dimers and recognizes the FG-repeats like the wild-type protein.

OK

2. The EGFP-NTF2 fluorescence intensity was used “as an indicator for the NPC middle location”. This peak is at ca. -24 nm from the NPC middle plane position i.e., towards the nuclear basket, based on Mab414 antibody staining, which is known to recognize Nup62. However, it was previously reported that Nup62 is localized at ca. +10 nm from the midplane towards the cytoplasmic side. Hence, the distance between the peak for eGFP-NTF2 here and peak for the Mab414 is ca. 34 nm.

Q: Still, it is not clear how the authors defined the position of Nup62 in the NPC using Mab414. From Fig. 2a/b in Schwarz-Herion 2007, it is the C-terminus of Nup62 that is localized at +10 nm from the central plane whereas the N-terminus has more of a bimodal distribution at ca. +17 and -20 nm (Fig. 2c/d). Yet, Mab414 should only recognize the FG domain of Nup62, which is located at the N-terminus. Can the authors please clarify?

We thank the reviewer for raising this point. Indeed, the precise localization of p62 – especially by mAb414 – is not straightforward. Therefore, we reconsidered the

literature and repeated our calibration based on an EM analysis of mAb414 itself without referring to Nup62, namely the publication by Grote et al., 1995 (M. Grote, U. Kubitscheck, R. Reichelt and R. Peters. Mapping of nucleoporins to the center of the nuclear pore complex by post-embedding immunogold electron microscopy. *Journal of Cell Science* 108, 2963–2972, 1995). In this publication the localization of binding sites specifically of mAb414 (and an anti-p62 antibody) was examined in 3T3 cells, human keratinocytes and rat liver cells. One result was that the binding site distributions were comparable. In their figure 6 they show the binding of mAb414 to NPCs in rat liver cells. We re-evaluated these data with regard to the position of the maximum of the binding site distribution and yielded the following result:

Binding site distribution of mAb414 along the axis of the NPC ($z=0$ corresponds to the central plane of the NPC, negative numbers refer to nucleoplasmic positions) according to Fig. 6g and 6h of Grote et al., 2021.

The mean value of this distribution was 5 ± 25 nm, the median was 4 nm. We would like to stress that these values differed by the mean position of mAb414 used in our work so far only by 5 nm and thus the difference does not affect any of our conclusions. Nevertheless, we used the value of 5 nm for our data analysis, modified Figs. 4B and 5B accordingly and reformulated the respective sections in the manuscript as follows:

Thank you for this analysis. However, the plot of “Count vs. Z” shown here is not taken directly from Grote 1995, nor is it apparent how it relates to Grote 1995. Noting that U.Kb. is a second author of Grote et al., might this be a re-analysis of data from that time? Can the authors please clarify?

If so, the authors might consider including the plot in the supplementary data or seek guidance from the Editor.

In the manuscript on page 10:

The location of the maximum NTF2 fluorescence was related to the NPC topology using mAb414 labelling Nup 62 and various other the nucleoporins (Davis and Blobel, 1986) as reference (supplemental Fig. 9 and Methods). Thereby, we found that NTF2

binds preferentially between 0 and 70 nm off the central NPC plane with its maximum at 29 nm towards the nuclear basket.

Not Supp Fig. S9 but Supp. Fig. S10.

In the Methods on page 11:

To determine the positions of maximum NTF2 and mAb414 accumulation, in both channels the position of the same NPC was selected manually and more precisely localized by four Gaussian fits in 45° angles. The maxima of the fits were averaged and the distance between the maxima of the NTF2 and mAb414 calculated.

We also changed the legend to supplemental Fig. 9 accordingly.

Supp. Fig. S10

Q: Mab414 ought to interact with other FG Nups besides Nup62. How do the authors know with certainty that only Nup62 was targeted?

The reviewer is correct that mAb414 does not only bind Nup62. Therefore, we now used EM-based data of the binding of mAb414 (Grote et al., 1995) as a reference for determining the NTF2 position as outlined in the answer to the previous question.

OK

3. Pg 15 and Fig. 5. “Our measured distribution of trajectory positions, although less well resolved than the EM data, showed a clear, single maximum in the central region of the NPC shifted towards the cytoplasm.”

Q: However, in Fig. 5B the maximum seems to be clearly in the region of the nuclear basket. Can the authors please comment on this or clarify?

We are very sorry. The reference on page 15 given in this context must be to Fig. 4B and not to Fig. 5. We corrected this. Fig. 4B shows the mentioned result.

OK

4. The authors state “Ribbeck and Görlich (2002) found that the transport rates of bigger transport cargoes scale with the number of loaded transport receptors” then speculate that each pre-60S particle is accompanied by one copy of CRM1 and more than one copy of Exp5.

Q: It may be good if the authors can expand their discussion on this point. Could the observed modes of behaviour (Fig. 3G) be influenced by the number of bound NTRs, especially Exp5? See also Paci eLIFE 2020 and Tu EMBOJ 2013.

We followed this suggestion of the reviewer and now wrote on page 14: “Ribbeck and Görlich (2002) found that the transport rates of bigger transport cargoes scale with the number of loaded transport receptors⁴⁶. This was later confirmed by studies of

nuclear import of exceptionally big artificial transport substrates in permeabilized mammalian cells⁴⁷⁻⁴⁹. Lowe et al. (2010) used quantum dots (QD) functionalized with importin β binding domains featuring a hydrodynamic radius of ~ 18 nm⁴⁷. Reducing the number of bound importin β molecules dramatically increased the QD dwell time in the central NPC channel⁴⁷. Tu et al. (2013) engineered β -galactosidase (β -gal) to carry four M9-signals resulting in an import cargo of comparable size ($18 \times 15 \times 9$ nm³). Their single molecule analysis revealed that loading the β -gal with one receptor allowed NPC binding but no translocation. For maximum import efficiency loading of four transportins per β -gal was necessary⁴⁸. In a more recent study⁴⁹ the import of artificial virus capsids with increasing diameters (17 to 36 nm) and varying numbers of NLS (zero to 240) was quantified in ensemble measurements. Again, the initial transport rate and efficiency scaled with the number of NLS/transport receptors attached to the capsids. Together, these studies illustrate that the enthalpic and entropic costs of “solving” a large cargo in the dense FG network can be compensated by multiple NTR-FG interactions.”

OK

5. What is the difference between mean translocation time, export duration, and dwell time?

Mean translocation time corresponds to the mean value of all dwell times. Dwell time corresponds to the interaction duration of single exported particles according to our definition given on page 8. Expectation value for the export duration: time constant determined by fitting an exponential decay function to the data of Fig. 4A indicated by the blue line in this figure. In the legend to Fig. 4A we wrote erroneously “export durations” where it should read “dwell times”. This was corrected. Probably this mistake contributed to a misunderstanding of the various designations.

OK

6. Refs 52 and 55 are replicates.

Corrected.

OK

The technology on display is impressive, and the scientific insights are novel, significant and timely. Hence, I would support its publication in Nature Communications once the above questions are clarified.

We thank the reviewer for this positive assessment of our work and hope that we addressed all concerns adequately.

Other points:

Fig. S3D: It is indicated that the sucrose gradient has a range from 15 to 45 %, but in the legend it is written 10-45 %.

We would like to thank again the reviewers for their positive assessment. We have addressed the remaining questions and suggestions of the third reviewer as follows (my answers are given in “green”):

Reviewer #3 (Remarks to the Author):

Stability of the EGFP-NTF2 pool at the NPC

The binding of EGFP-NTF2 to the NPC in permeabilized cells is short-lived with a binding time of 5.8 ms (Kubitscheck et al., 2005), but probably more complex in molecularly crowded surroundings (Wagner et al., 2015). To address the situation in living cells we measured the dynamics of eGFP-NTF2 at the nuclear envelope using FRAP. We observed a rapid recovery of the bleached eGFP-NTF2 at the nuclear envelope with an estimated half-time of 0.2 to 0.4 ms. The mobile fraction was roughly 90%. This demonstrated a rapid exchange of the fluorescent label and should result in an prolonged localization precision of NPC (see below).

Typo: In Supp Fig. 4B, it is written “The estimated recovery time is 0.2 to 0.4 s”.

We are very sorry; the typo was in our rebuttal letter: we wrote mistakenly ms here instead of s. The images were taken with a frame rate of 90 ms/frame. The figure legend is correct.

What are the levels of EGFP-NTF2 in comparison to endogenous NTF2?

To answer this question we now included Western Blots of the cell lysate using antibodies against hNTF2 and GFP (supplemental Figure 4A). GFP-tagged NTF2 is expressed at a similar level like the endogenous NTF2.

OK. However, a loading control is missing in Supp. Fig. 4A.

In order to create the shown gel, we took (2x) 5 μ L of a full cell lysate and applied it to the two gel columns. Since we used the same lysate, there is no further gel loading control possible or required. In order to make this clear we added the sentence: “5 μ L of the full lysate were applied to each of the gel columns.” to the legend of Supp. Fig. 4a.

Q: Still, it is not clear how the authors defined the position of Nup62 in the NPC using Mab414. From Fig. 2a/b in Schwarz-Herion 2007, it is the C-terminus of Nup62 that is localized at +10 nm from the central plane whereas the N-terminus has more of a bimodal distribution at ca. +17 and -20 nm (Fig. 2c/d). Yet, Mab414 should only recognize the FG domain of Nup62, which is located at the N-terminus. Can the authors please clarify?

We thank the reviewer for raising this point. Indeed, the precise localization of p62 – especially by mAb414 – is not straightforward. Therefore, we reconsidered the literature and repeated our calibration based on an EM analysis of mAb414 itself without referring to Nup62, namely the publication by Grote et al., 1995 (M. Grote, U. Kubitscheck, R. Reichelt and R. Peters. Mapping of nucleoporins to the center of the nuclear pore complex by post-embedding immunogold electron microscopy. *Journal*

of *Cell Science* 108, 2963–2972, 1995). In this publication the localization of binding sites specifically of mAb414 (and an anti-p62 antibody) was examined in 3T3 cells, human keratinocytes and rat liver cells. One result was that the binding site distributions were comparable. In their figure 6 they show the binding of mAb414 to NPCs in rat liver cells. We re-evaluated these data with regard to the position of the maximum of the binding site distribution and yielded the following result:

Binding site distribution of mAb414 along the axis of the NPC ($z=0$ corresponds to the central plane of the NPC, negative numbers refer to nucleoplasmic positions) according to Fig. 6g and 6h of Grote et al., 2021.

The mean value of this distribution was 5 ± 25 nm, the median was 4 nm. We would like to stress that these values differed by the mean position of mAb414 used in our work so far only by 5 nm and thus the difference does not affect any of our conclusions. Nevertheless, we used the value of 5 nm for our data analysis, modified Figs. 4B and 5B accordingly and reformulated the respective sections in the manuscript as follows:

Thank you for this analysis. However, the plot of “Count vs. Z” shown here is not taken directly from Grote 1995, nor is it apparent how it relates to Grote 1995. Noting that U.Kb. is a second author of Grote et al., might this be a re-analysis of data from that time? Can the authors please clarify?

If so, the authors might consider including the plot in the supplementary data or seek guidance from the Editor.

Indeed, the graph was not shown in this format in Grote et al., 1995. To determine the maximum of the binding site distribution of mAb414 we determined the binding positions of the mAb414 in Fig. 6g of Grote et al., 1995 and plotted the z-coordinates (i.e., the direction along the transport axis). This resulted in the shown plot. To explain this approach, we now write in the legend to Supp. Fig. 10: “The final distances were corrected by 5 nm to account for the mAb414 binding site maximum along the transport axis (shifted towards the cytoplasm with regard to the central plane of the NPC) as determined from localizations of mAb414 labeled by gold-anti-mouse antibodies in ultrathin sections of rat liver tissue by transmission electron microscopy (Fig. 6g in Grote et al., 1995^{3,4}) ...”

In the manuscript on page 10:

The location of the maximum NTF2 fluorescence was related to the NPC topology using mAb414 labelling Nup 62 and various other the nucleoporins (Davis and Blobel, 1986) as reference (supplemental Fig. 9 and Methods). Thereby, we found that NTF2 binds preferentially between 0 and 70 nm off the central NPC plane with its maximum at 29 nm towards the nuclear basket.

Not Supp Fig. S9 but Supp. Fig. S10.

Correct. As above, the mistake was in the rebuttal letter.

In the Methods on page 11:

To determine the positions of maximum NTF2 and mAb414 accumulation, in both channels the position of the same NPC was selected manually and more precisely localized by four Gaussian fits in 45° angles. The maxima of the fits were averaged and the distance between the maxima of the NTF2 and mAb414 calculated.

We also changed the legend to supplemental Fig. 9 accordingly.

Supp. Fig. S10

Correct. As above, the mistake was in the rebuttal letter.

Fig. S3D: It is indicated that the sucrose gradient has a range from 15 to 45 %, but in the legend it is written 10-45 %.

Amended. 10-45 % is the correct range.